# AdaIR: Adaptive All-in-One Image Restoration via Frequency Mining and Modulation

**Yuning Cui**[1], **Syed Waqas Zamir**[2], **Salman Khan**[3,4],
**Alois Knoll**[1], **Mubarak Shah**[5], **Fahad Shahbaz Khan**[3,6]
[1]Technical University of Munich
[2]Inception Institute of Artificial Intelligence
[3]Mohammed Bin Zayed University of AI
[4]Australian National University
[5]University of Central Florida
[6]Linköping University

## Abstract

In the image acquisition process, various forms of degradation, including noise, blur, haze, and rain, are frequently introduced. These degradations typically arise from the inherent limitations of cameras or unfavorable ambient conditions. To recover clean images from their degraded versions, numerous specialized restoration methods have been developed, each targeting a specific type of degradation. Recently, all-in-one algorithms have garnered significant attention by addressing different types of degradations within a *single model* without requiring the prior information of the input degradation type. However, most methods purely operate in the spatial domain and do not delve into the distinct frequency variations inherent to different degradation types. To address this gap, we propose an adaptive all-in-one image restoration network based on frequency mining and modulation. Our approach is motivated by the observation that different degradation types impact the image content on different frequency subbands, thereby requiring different treatments for each restoration task. Specifically, we first mine low- and high-frequency information from the input features, guided by the adaptively decoupled spectra of the degraded image. The extracted features are then modulated by a bidirectional operator to facilitate interactions between different frequency components. Finally, the modulated features are merged into the original input for a progressively guided restoration. With this approach, the model achieves adaptive reconstruction by accentuating the informative frequency subbands according to different input degradations. Extensive experiments demonstrate that the proposed method, AdaIR, achieves state-of-the-art performance on different image restoration tasks, including image denoising, dehazing, deraining, motion deblurring, and low-light image enhancement. The code is available at `https://github.com/c-yn/AdaIR`.

## 1 Introduction

Image restoration is the task of generating a clean image by removing degradations (*e.g.*, noise, haze, blur, rain) from the original input (Zamir et al., 2021; Cui et al., 2024c). It serves as a vital component in numerous downstream applications across diverse domains, including image/video content creation, surveillance, medical imaging, and remote sensing. Given its inherently ill-posed nature, effective image restoration demands learning strong image priors from large-scale data. To this end, deep neural network-based image restoration approaches (Zamir et al., 2020a; Tsai et al., 2022b; Cui et al., 2023c;e; Wang et al., 2024b;a; Liu et al., 2025) have emerged as preferable choices over the conventional handcrafted algorithms (He et al., 2010; Kim & Kwon, 2010; Michaeli & Irani, 2013). Deep-learning methods learn image priors either implicitly from data (Ren et al., 2021; Nah et al., 2022; Dong et al., 2020a), or explicitly by incorporating task-specific knowledge into the network architectures (Tu et al., 2022; Wang et al., 2022; Zamir et al., 2022a; 2020b; Cui et al., 2024b). Despite promising results on individual restoration tasks, these approaches are either not generalizable beyond the specific degradation types and levels which hinders their broader application, or require training separate copies of the same network on different degradation types, which is computationally expensive and tedious procedure, and maybe infeasible solution for deployment on resource-constraint edge-devices. Therefore, there is a need to develop an all-in-one image restoration

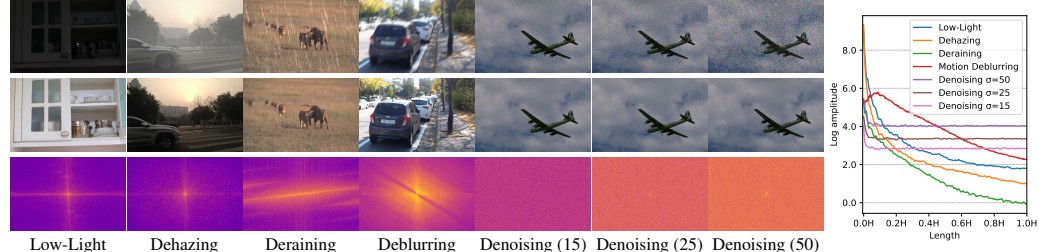

| Low-Light | Dehazing | Deraining | Deblurring | Denoising (15) | Denoising (25) | Denoising (50) |

Figure 1: *Left*, from top to bottom: degraded images, ground truth, and Fourier spectra of residual images obtained by subtracting the degraded images from the ground truth. The images are obtained from LOL-v1 (Wei et al., 2018), SOTS (Li et al., 2018), Rain100L (Li et al., 2018), GoPro (Nah et al., 2017), and BSD68 (Martin et al., 2001) with different noise factors, respectively. *Right*, the sub-graph illustrates the mean values of Fourier spectra on the square of length shown on the x-axis, across five tasks. The spectra are all resized to $320 \times 320$ for comparisons. As seen, different tasks pay different attention to different frequency subbands. For example, there are larger discrepancies in low frequency between degraded and target pairs of the low-light image enhancement and dehazing datasets. In contrast, the frequency differences are generally evenly distributed for image denoising.

method that can handle images with different degradation types, without requiring prior information regarding the corruption present in the input images.

Recently, an increasing number of attempts have been made (Ma et al., 2023; Shi et al., 2024; Gao et al., 2023; Cui et al., 2024d) to address multiple degradations with a single model. These include using a degradation-aware encoder in the restoration network learned via contrastive learning paradigm (Li et al., 2022); designing a two-stage framework IDR (Zhang et al., 2023), where the first stage is dedicated to task-oriented knowledge collection based on underlying physics charac-

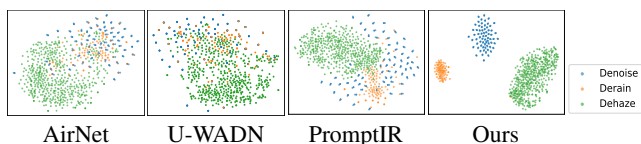

| AirNet | U-WADN | PromptIR | Ours |

Figure 2: The t-SNE results of intermediate features produced by the three-task all-in-one models. Our model is better at learning discriminative degradation contexts.

teristics of degradation types, and the second stage is responsible for ingredients-oriented knowledge integration that progressively restores the image; or developing prompt-learning strategies (Potlapalli et al., 2023; Ma et al., 2023) inspired from their success in the natural language processing (Brown et al., 2020). Nonetheless, most existing approaches purely operate in the spatial domain and do not consider frequency information. However, as shown in Fig. 1, we observe that different degradations may impact the image content on different frequency subbands. For instance, on the one hand, noisy and rainy images are contaminated with high-frequency content, while on the other hand, low-light and hazy images are dominated by low-frequency degraded content, thus indicating the need to treat each restoration task on its own merits.

In this paper, we propose an adaptive all-in-one image restoration framework based on frequency mining and modulation. Specifically, the frequency mining module extracts different frequency signals from the input features, guided by an adaptive spectra decomposition of the degraded input image. The extracted features are then refined using a bidirectional module, which facilitates the interactions between different frequency components by exchanging complementary information. Finally, these modulated features are used to transform the original input features via an efficient transposed cross-attention mechanism. With the proposed key design choices, our method can learn discriminative degradation context more effectively than other competing approaches, as shown in Fig. 2. Overall, the following are the main contributions of our work.

- We propose an adaptive all-in-one image restoration framework that leverages both spatial and frequency domain information to effectively decouple degradations from the desired clean image content.

- We introduce the Adaptive Frequency Learning Block (AFLB), which is a plugin block specifically designed for easy integration into existing image restoration architectures. The AFLB performs two sequential tasks: firstly, through its Frequency Mining Module (FMiM), it generates low- and high-frequency feature maps via guidance obtained from the spectra

decomposition of the original degraded image; secondly, the Frequency Modulation Module (FMoM) within the AFLB calibrates these features by enabling the exchange of information across different frequency bands to effectively handle diverse types of image degradations.

- Extensive experiments demonstrate that our AdaIR algorithm sets new state-of-the-art performance on several all-in-one image restoration tasks, including image denoising, dehazing, deraining, motion deblurring, and low-light image enhancement.

## 2 RELATED WORK

**Single-Task Image Restoration.** Image restoration aims to reconstruct a clean image from its degraded counterpart. Since it is a highly ill-posed problem, many conventional methods have been proposed that utilize hand-crafted features to reduce the solution space (Berman et al., 2016; He et al., 2010). Such solutions, though perform well on some datasets, may not generalize well to complicated real-world images (Zhang et al., 2022). Recently, with the rapid advancements in deep learning, a great number of convolutional neural network (CNN) based methods have been proposed and attained superior performance over traditional methods on various image restoration tasks, such as image denoising (Zhang et al., 2017a; 2018), dehazing (Qin et al., 2020; Cui et al., 2024a), deraining (Jiang et al., 2020; Ren et al., 2019), and motion deblurring (Cho et al., 2021; Cui et al., 2023f). For example, the recent UHDVD obtains one of the best reconstruction accuracies when tested on UHD blurry frames (Ren et al., 2024). To model long-range dependencies, Transformer models have been introduced to low-level tasks and significantly advanced state-of-the-art performance (Guo et al., 2022; Tsai et al., 2022a). Despite the obtained promising performance, these task-specific methods lack generalization beyond certain degradation types and levels. For general image restoration, several network design-based approaches are proposed, which perform favorably on different restoration tasks (Wang et al., 2022; Liang et al., 2021; Zamir et al., 2022a). Although these networks demonstrate robust performance on various restoration tasks, they require training separate copies on different datasets and tasks. Furthermore, applying a separate model for each possible degradation is resource-intensive, and often impractical for deployment, especially on edge devices.

**All-in-One Image Restoration.** All-in-one image restoration methods address numerous degradations within a single model (Yang et al., 2024; Jiang et al., 2023; Chen & Pei, 2023). Early unified models (Chen et al., 2021b; Li et al., 2020) employ distinct encoder and decoder heads to attend to different restoration tasks. However, these non-blind methods need prior knowledge about the degradation involved in the corrupted image in order to channelize it to the relevant restoration head. To achieve blind all-in-one restoration, AirNet (Li et al., 2022) learns the degradation representation from the corrupted images using contrastive learning, and the learned representation is then used to restore the clean image. The subsequent method, IDR (Zhang et al., 2023), models different degradations depending on the underlying physics principles and achieves all-in-one image restoration in two stages. Zhu *et al.* (Zhu et al., 2023) formulates an efficient unified model by learning weather-general and weather-specific features in two stages. Recently, several prompt-learning-based schemes have been proposed (Ma et al., 2023; Conde et al., 2024; Ai et al., 2024). For instance, PromptIR (Potlapalli et al., 2023) presets a series of tunable prompts to encode discriminative information about degradation types, which involve a large number of parameters. Different from most methods, which operate only in the spatial domain (Park et al., 2023), this paper presents an all-in-one image restoration algorithm that exploits information both in spatial and frequency domains.

**Frequency Networks.** Frequency processing has become a prevalent technique in the field of image restoration. For example, several works (Zhou et al., 2024; Cui et al., 2023d;b) employ adaptive convolutions and softmax mechanisms to decouple features. However, these methods operate exclusively in the spatial domain, limiting their ability to capture a broad spectrum of frequencies and diminishing their effectiveness in frequency learning. Furthermore, their use of concatenation or channel attention for frequency interactions fails to exploit the unique properties of different frequency bands. Other approaches (Kong et al., 2023; Mao et al., 2023; 2024) leverage frequency transformation techniques, such as Fourier or Wavelet transforms, to map spatial features into the frequency domain, followed by convolutions or learnable parameters for spectral refinement. However, these methods lack explicit frequency interactions, and their parameters remain fixed after training, hindering adaptability to diverse degradation types. Zheng *et al* (Zheng et al., 2021) employ a deep CNN block to learn bandpass filters for image demoireing. In the context of all-in-one image restoration, a few methods (Gao et al., 2023; Shi et al., 2024) employ manual or non-adaptive approaches for feature separation and execute frequency interactions without accounting for the distinct characteristics of different frequency components. Unlike the above algorithms, our

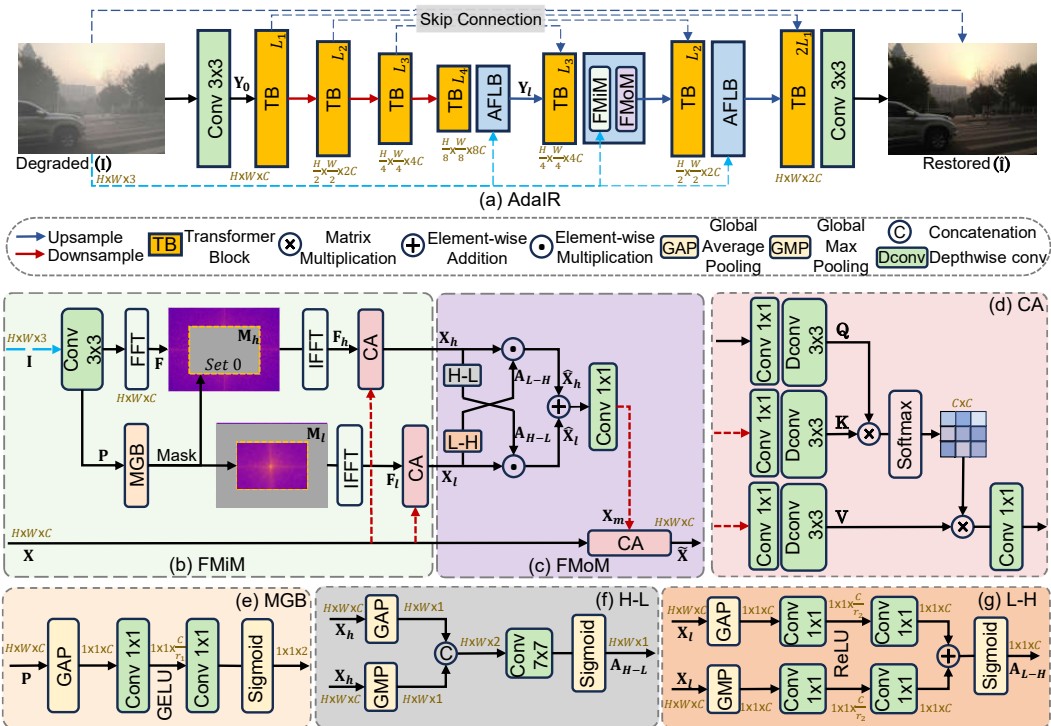

Figure 3: (a) The overall pipeline of AdaIR. It is a Transformer-based encoder-decoder architecture, employing TB (Zamir et al., 2022a) and a novel Adaptive Frequency Learning Blocks (AFLB). Each AFLB contains (b) Frequency Mining Module (FMiM) that extracts different frequency components from input features guided by the adaptively decoupled spectra of the degraded input image, and (c) Frequency Modulation Module (FMoM) that exchanges the complementary information between different frequency features. (d) Cross Attention (CA) (Zamir et al., 2022a). (e) Mask Generation Block (MGB) that yields a frequency boundary for spectra decomposition. (f) H-L unit (Woo et al., 2018) delivers high-frequency attention maps to enrich Low-frequency features. (g) L-H unit enhances high-frequency features by complementing them with low-frequency features. FFT and IFFT denote the Fast Fourier Transform and its inverse operator, respectively.

approach explicitly operates in the frequency domain and realizes adaptability to various degradations. Furthermore, we employ distinct attention mechanisms to facilitate frequency interactions, leveraging the unique characteristics of different frequency bands to enable more effective frequency learning.

## 3 METHOD

### 3.1 OVERALL PIPELINE

Fig. 3 presents the pipeline of AdaIR. The overall goal of our AdaIR framework is to learn a unified model $\mathbf{M}$ that can recover a clean image $\hat{\mathbf{I}}$ from a degraded image $\mathbf{I}$, without any prior information of degradation type $\mathbf{D}$ present in the input image $\mathbf{I}$. Formally, given a degraded image $\mathbf{I} \in \mathbb{R}^{H \times W \times 3}$, AdaIR first extracts shallow features $\mathbf{Y_0} \in \mathbb{R}^{H \times W \times C}$ using a $3 \times 3$ convolution; where $H \times W$ denotes the spatial size and $C$ represents the number of channels. Next, these features $\mathbf{Y_0}$ are processed through a 4-level encoder-decoder network. Each level of the encoder employs multiple Transformer blocks (TBs) (Zamir et al., 2022a), where the number of blocks gradually increases from the top level to the bottom level, facilitating a computationally efficient design. The encoder takes high-resolution features $\mathbf{Y_0}$ as input, and progressively transforms them into a lower-resolution latent representation $\mathbf{Y}_l \in \mathbb{R}^{\frac{H}{8} \times \frac{W}{8} \times 8C}$. On the decoder side, the latent features $\mathbf{Y}_l$ are processed with interleaved Adaptive Frequency Learning Block (AFLB) and TBs to progressively reconstruct high-resolution clean output. Particularly, between every two levels of the decoder, we insert the AFLB that adaptively segregates the degradation content from the clean image content in the frequency domain, and subsequently assists in refining features in the spatial domain for effective image restoration.

Since different types of degradations affect image content at different frequency bands (as shown in Fig. 1), we specifically design the Adaptive Frequency Learning Block (AFLB) that extracts low- and high-frequency components from the input features and then modulate them to accentuate the corresponding informative subbands for each degradation. Next, we describe the two key components of AFLB: (1) **F**requency **Mi**ning **M**odule (FMiM) and **F**requency **Mo**dulation **M**odule (FMoM).

## 3.2 FREQUENCY MINING MODULE (FMiM)

As shown in Fig. 3(b), given as inputs both the degraded image $\mathbf{I}$ and the intermediate features $\mathbf{X} \in \mathbb{R}^{H \times W \times C}$, FMiM mines different frequency representations from $\mathbf{X}$ with the guidance of adaptively decoupled spectra of $\mathbf{I}$. Primarily, FMiM consists of three steps, *i.e.,*, domain transformation, mask generation, and feature extraction.

For the domain transformation, FMiM applies a $3 \times 3$ convolution on the degraded image $\mathbf{I}$ to expand the channel capacity to align with that of the input features $\mathbf{X}$. These output features are transformed into spectral domain representation $\mathbf{F} \in \mathbb{R}^{H \times W \times C}$ via the Fast Fourier Transform (FFT).

Since we want to adaptively extract different frequency parts from the input features $\mathbf{X}$, we design a lightweight Mask Generation Block (MGB) to generate a 2D mask that serves as a frequency boundary to separate the spectra of input image $\mathbf{I}$. The cutoff frequency boundary adaptively changes according to the type of degradation present in the image. As illustrated in Fig. 3(e), the projected feature map $\mathbf{P}$ is first mapped into a vector using a global average pooling operator and then passes through two $1 \times 1$ convolution layers with the GELU activation function in between to produce two factors ranging from 0 to 1, which define the mask size by multiplying with the width and height of the spectra. The mask generation process can be formally expressed as:

$$[\alpha, \beta] = \delta \left( W_2^{1 \times 1} \left( \sigma \left( W_1^{1 \times 1} \left( \text{GAP}_s \left( \mathbf{P} \right) \right) \right) \right) \right) \tag{1}$$

where $\text{GAP}_s$ denotes spatial global average pooling, $\sigma$ is the GELU activation, and $\delta$ indicates the Sigmoid function. The convolution $W_1$ and $W_2$ have the reduction ratios of $r_1$ and $\frac{C}{2r_1}$, respectively, progressively downsampling the channel dimensions to 2. Subsequently, the binary mask $\mathbf{M}_l \in \{0, 1\}^{H \times W}$ for extracting low frequency can be obtained by setting $\mathbf{M}_l[\frac{H}{2} - \alpha \frac{H}{k} : \frac{H}{2} + \alpha \frac{H}{k}, \frac{W}{2} - \beta \frac{W}{k} : \frac{W}{2} + \beta \frac{W}{k}] = 1$, where $k$ is set to a value of 128, as the curve junction is relatively small in Fig. 1. Accordingly, the mask for high frequency $\mathbf{M}_h$ is obtained by setting the values within the remaining region as 1. Subsequently, we can obtain the adaptively decoupled features by applying the learned masks to the spectra via element-wise multiplication and using the inverse Fourier transform.

Next, we adapt the multi-dconv head transposed cross attention (Fig. 3(d)) (Zamir et al., 2022a; Chen et al., 2021a) to mine the different feature parts from the input features with the guidance of $\mathbf{F}_l$ and $\mathbf{F}_h$. Overall, the feature extraction process is defined as:

$$\mathbf{X}_* = \text{softmax} \left( \mathbf{Q} \mathbf{K}^\top / \alpha \right) \mathbf{V}, \qquad \text{where,} \tag{2}$$

$$\mathbf{Q} = DW_1 \left( W_3^{1 \times 1} (\mathbf{F}_*) \right), \mathbf{K} = DW_2 \left( W_4^{1 \times 1} (\mathbf{X}) \right), \mathbf{V} = DW_3 \left( W_5^{1 \times 1} (\mathbf{X}) \right), \text{where,} \tag{3}$$

$$\mathbf{F}_* = \mathcal{F}^{-1} \left( \mathbf{M}_* \odot \mathbf{F} \right), \tag{4}$$

where $* \in \{l, h\}$ is an indicator for low/high frequency, $DW$ represents a $3 \times 3$ depth-wise convolution, $\odot$ is element-wise multiplication, $\mathcal{F}^{-1}$ indicates the inverse fast Fourier transform, $\mathbf{Q}$, $\mathbf{K}$ and $\mathbf{V}$ are *query*, *key* and *value* projections, respectively, which are separately generated with a sequential application of $1 \times 1$ convolution and $3 \times 3$ depth-wise convolution, and $\alpha$ is a learnable scaling factor to control the magnitude of the dot product result of $\mathbf{Q}$ and $\mathbf{K}$ before using the softmax function.

## 3.3 FREQUENCY MODULATION MODULE (FMoM)

We devise FMoM to facilitate the cross interaction between low-frequency mined features and high-frequency mined features (see Fig. 3(c)). The goal is to cross complement one type of mined features with the other. For instance, high-frequency features contain edges and fine texture details, and thus we use this information to enrich low-frequency mined features via a super-lightweight spatial attention unit (H-L) (Fig. 3(f)). Similarly, the global information present in low-frequency features is passed to the high-frequency branch through the channel attention unit (L-H), illustrated in Fig. 3(g).

**H-L Unit:** This unit computes the spatial attention map from high-frequency mined features that are used to complement features of the low-frequency branch. The H-L unit (Woo et al., 2018) uses two different channel-wise pooling techniques in parallel to produce two single-channel spatial feature

maps, each of size $H \times W \times 1$. These maps are then concatenated along the channel dimension. The concatenated features are further refined with a $7 \times 7$ convolution, followed by a sigmoid operation to generate the final spatial attention map, which is then used to obtain the modulated low-frequency features via element-wise multiplication. Overall, the process of the H-L Unit is given by:

$$\hat{\mathbf{X}}_l = \mathbf{X}_l \odot \mathbf{A}_{H-L}, \qquad \text{where,} \tag{5}$$

$$\mathbf{A}_{H-L} = \delta \left( W_6^{7 \times 7}([\text{GAP}_c(\mathbf{X}_h), \text{GMP}_c(\mathbf{X}_h)]) \right), \tag{6}$$

where $\mathbf{W}_6$ has a channel reduction ratio of 2. $\delta$ is the sigmoid function. $\text{GAP}_c$ and $\text{GMP}_c$ are the channel-wise global average pooling and max pooling, respectively. $[\cdot, \cdot]$ indicates concatenation.

**L-H Unit:** It is a dual branch module that processes incoming low-frequency mined features, yielding a feature descriptor that is subsequently used to attend to the high-frequency mined features. Specifically, given the mined low-frequency features $\mathbf{X}_l \in \mathbb{R}^{H \times W \times C}$, the top branch of the L-H unit applies global average pooling along spatial dimension to obtain a feature vector of size $1 \times 1 \times C$, followed by two convolutional layers with the ReLU activation function in between. The bottom branch of the L-H unit employs the same structure, with the only difference of Max pooling at the head. The results of the two branches are added together, on which the sigmoid function is applied to produce the final attention descriptor $\mathbf{A}_{L-H} \in \mathbb{R}^{1 \times 1 \times C}$, which is used to modulate the mined high-frequency features $\mathbf{X}_h$. The process of the L-H Unit is expressed by:

$$\hat{\mathbf{X}}_h = \mathbf{X}_h \odot \mathbf{A}_{L-H}, \qquad \text{where,} \tag{7}$$

$$\mathbf{A}_{L-H} = \delta \left( W_8^{1 \times 1} \left( \gamma \left( W_7^{1 \times 1}(\text{GAP}_s(\mathbf{X}_l)) \right) \right) + W_{10}^{1 \times 1} \left( \gamma(W_9^{1 \times 1}(\text{GMP}_s(\mathbf{X}_l))) \right) \right), \tag{8}$$

where $\delta$ is the sigmoid function, $\hat{\mathbf{X}}_h$ is the modulated high-frequency features, $\text{GAP}_s$ and $\text{GMP}_s$ represent the global average pooling and max pooling along the spatial dimensions, respectively. $\gamma$ indicates the ReLU activation function. $\mathbf{W}_7$ and $\mathbf{W}_9$ have a reduction ratio of $r_2$ for the channel adjustment, while $\mathbf{W}_8$ and $\mathbf{W}_{10}$ have an increasing ratio of $r_2$. The parameters are shared among $\mathbf{W}_7$ and $\mathbf{W}_9$, $\mathbf{W}_8$ and $\mathbf{W}_{10}$ for computational efficiency.

Subsequently, the modulated high-frequency features $\hat{\mathbf{X}}_h$ and low-frequency features $\hat{\mathbf{X}}_l$ are aggregated and processed via a $1 \times 1$ convolution to obtain $\mathbf{X}_m$, which is merged into the original input $\mathbf{X}$ using the cross-attention unit, where the *query* $\mathbf{Q}$ tensor is produced from $\mathbf{X}$ while $\mathbf{X}_m$ yields the *key* $\mathbf{K}$ and *value* $\mathbf{V}$ tensors. By using FMiM and FMoM, the high-frequency and low-frequency contents of the input features are separately and adaptively modulated according to the degradation type present in the corrupted input image, leading to adaptive all-in-one image restoration.

## 4 EXPERIMENTS

To validate the efficacy of the proposed AdaIR, we conduct experiments by strictly following previous state-of-the-art works (Potlapalli et al., 2023; Li et al., 2022) under two different settings: **(1) All-in-One**, and **(2) Single-task**. In the All-in-One setting, a unified model is trained to perform image restoration across multiple degradation types. Whereas, within the Single-task setting, separate models are trained for each specific restoration task. We provide single-task results, additional ablation experiments, visual examples, and more details on the architecture in the Appendix. In tables, the best and second-best image fidelity scores (PSNR and SSIM (Wang et al., 2004)) are highlighted in red and blue, respectively.

**Implementation Details.** Our AdaIR presents an end-to-end trainable solution without the necessity for pretraining any individual component. The architecture of AdaIR employs a 4-level encoder-decoder structure, with varying numbers of Transformer blocks (TB) at each level, specifically [4, 6, 6, 8] from level-1 to level-4. We integrate one AFLB block between every two consecutive decoder levels, amounting to a total of three AFLBs in the overall network.

For training, we adopt a batch size of 32 in the all-in-one setting and a batch size of 8 in the single-task setting. The network optimization is achieved through an L1 loss function, employing the Adam optimizer ($\beta 1 = 0.9$ and $\beta 2 = 0.999$), with a learning rate of $2e^{-4}$, over the course of 150 epochs. During the training process, cropped patches sized at $128 \times 128$ pixels are provided as input, with additional augmentation applied via random horizontal and vertical flips. All experiments are conducted on NVIDIA Tesla A100 40G GPUs using PyTorch.

**Datasets.** In preparing datasets for training and testing, we closely follow prior works (Potlapalli et al., 2023; Li et al., 2022). For single-task image dehazing, we use one of the first standard dehazing

Table 1: Comparisons under the three-degradation all-in-one setting: a unified model is trained on a combined set of images obtained from all degradation types and levels. On Rain100L (Yang et al., 2019) for image deraining, AdaIR yields 0.7 dB gain over Art$_{PromptIR}$ (Wu et al., 2024).

| Method | Dehazing on SOTS | | Deraining on Rain100L | | Denoising on BSD68 | | | | | | Average | | Params |
| | | | | | $\sigma = 15$ | | $\sigma = 25$ | | $\sigma = 50$ | | | | |
| | PSNR | SSIM | PSNR | SSIM | PSNR | SSIM | PSNR | SSIM | PSNR | SSIM | PSNR | SSIM | |
| BRDNet (Tian et al., 2020) | 23.23 | 0.895 | 27.42 | 0.895 | 32.26 | 0.898 | 29.76 | 0.836 | 26.34 | 0.693 | 27.80 | 0.843 | 1.11M |
| LPNet (Gao et al., 2019) | 20.84 | 0.828 | 24.88 | 0.784 | 26.47 | 0.778 | 24.77 | 0.748 | 21.26 | 0.552 | 23.64 | 0.738 | 2.84M |
| FDGAN (Dong et al., 2020b) | 24.71 | 0.929 | 29.89 | 0.933 | 30.25 | 0.910 | 28.81 | 0.868 | 26.43 | 0.776 | 28.02 | 0.883 | - |
| MPRNet (Zamir et al., 2021) | 25.28 | 0.955 | 33.57 | 0.954 | 33.54 | 0.927 | 30.89 | 0.880 | 27.56 | 0.779 | 30.17 | 0.899 | 20.1M |
| DL (Fan et al., 2019) | 26.92 | 0.931 | 32.62 | 0.931 | 33.05 | 0.914 | 30.41 | 0.861 | 26.90 | 0.740 | 29.98 | 0.876 | 2.09M |
| AirNet (Li et al., 2022) | 27.94 | 0.962 | 34.90 | 0.968 | 33.92 | 0.933 | 31.26 | 0.888 | 28.00 | 0.797 | 31.20 | 0.910 | 8.93M |
| Restormer (Zamir et al., 2022a) | 27.78 | 0.958 | 33.78 | 0.958 | 33.72 | 0.930 | 30.67 | 0.865 | 27.63 | 0.792 | 30.75 | 0.901 | 26.13M |
| PromptIR (Potlapalli et al., 2023) | 30.58 | 0.974 | 36.37 | 0.972 | 33.98 | 0.933 | 31.31 | 0.888 | 28.06 | 0.799 | 32.06 | 0.913 | 32.96M |
| U-WADN (Xu et al., 2024) | 29.21 | 0.971 | 35.36 | 0.968 | 33.73 | 0.931 | 31.14 | 0.886 | 27.92 | 0.793 | 31.47 | 0.910 | 6M |
| Art$_{PromptIR}$ (Wu et al., 2024) | 30.83 | 0.979 | 37.94 | 0.982 | 34.06 | 0.934 | 31.42 | 0.891 | 28.14 | 0.801 | 32.49 | 0.917 | 33M |
| **AdaIR (Ours)** | 31.06 | 0.980 | 38.64 | 0.983 | 34.12 | 0.935 | 31.45 | 0.892 | 28.19 | 0.802 | 32.69 | 0.918 | 28.77M |

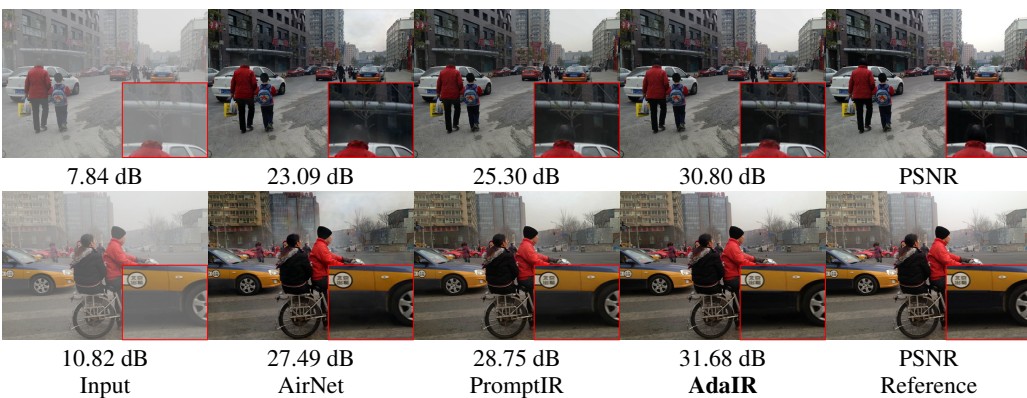

| 7.84 dB | 23.09 dB | 25.30 dB | 30.80 dB | PSNR |
| 10.82 dB | 27.49 dB | 28.75 dB | 31.68 dB | PSNR |
| Input | AirNet | PromptIR | **AdaIR** | Reference |

Figure 4: Image dehazing comparisons on SOTS (Li et al., 2018) between all-in-one methods. Compared to other algorithms, our method is more effective in haze removal.

datasets, SOTS (Li et al., 2018), which comprises 72,135 training images and 500 testing images. For single-task image deraining, we utilize Rain100L (Yang et al., 2019), which contains 200 clean-rainy image pairs for training and 100 pairs for testing. For single-task image denoising, we combine images of BSD400 (Arbelaez et al., 2010) and WED (Ma et al., 2016) datasets for model training; the BSD400 encompasses 400 training images, while the WED dataset consists of 4,744 images. Starting from these clean images of BSD400 (Arbelaez et al., 2010) and WED (Ma et al., 2016), we generate their corresponding noisy versions by adding Gaussian noise with varying levels ($\sigma \in \{15, 25, 50\}$). Denoising task evaluation is performed on BSD68 (Martin et al., 2001) and Urban100 (Huang et al., 2015). Finally, under the all-in-one setting, we train a single model on the combined set of the aforementioned training datasets and directly test it across multiple restoration tasks.

## 4.1 ALL-IN-ONE RESULTS: THREE DISTINCT DEGRADATIONS

We evaluate the performance of our *all-in-one* AdaIR on three different restoration tasks, including image dehazing, deraining, and denoising. We compare AdaIR against various general image restoration methods (BRDNet (Tian et al., 2020), LPNet (Gao et al., 2019), FDGAN (Dong et al., 2020b), MPRNet (Zamir et al., 2021), and Restormer (Zamir et al., 2022a)), as well as specialized all-in-one approaches (DL (Fan et al., 2019), AirNet (Li et al., 2022), PromptIR (Potlapalli et al., 2023), U-WADN (Xu et al., 2024), and Art$_{PromptIR}$ (Wu et al., 2024)). Table 1 shows that AdaIR provides consistent performance gains over the other competing approaches. When averaged across various restoration tasks and settings, our AdaIR obtains 0.2 dB PSNR gain over the recent best method Art$_{PromptIR}$ (Wu et al., 2024), and 0.63 dB improvement over the recent algorithm PromptIR (Potlapalli et al., 2023). Specifically, compared to Art$_{PromptIR}$ (Wu et al., 2024), AdaIR yields a substantial boost of 0.7 dB on the deraining task, and 0.23 dB on the dehazing task. We provide visual examples in Fig. 4 for dehazing, Fig. 5 for deraining, and Fig. 6 for denoising. These examples show that our AdaIR is effective in removing degradations, and generates images that are visually closer to the ground truth than those of the other approaches (Potlapalli et al., 2023; Li et al., 2022). Particularly, in the restored images, our method preserves better structural fidelity and fine textures.

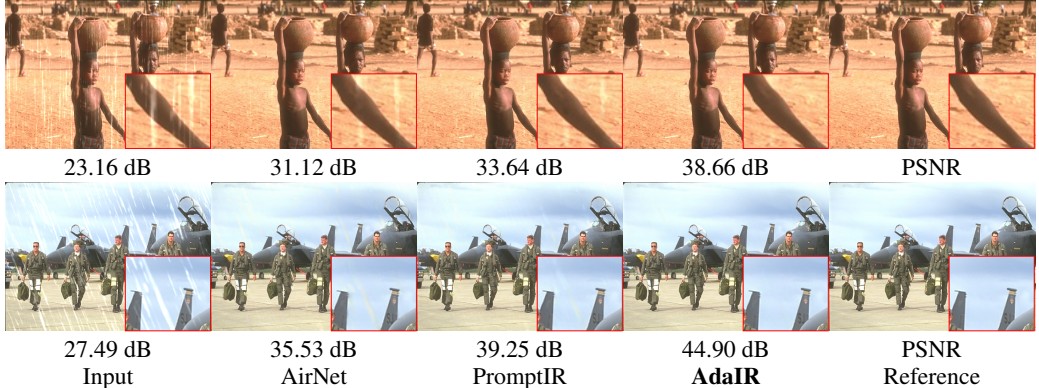

| 23.16 dB | 31.12 dB | 33.64 dB | 38.66 dB | PSNR |
| 27.49 dB | 35.53 dB | 39.25 dB | 44.90 dB | PSNR |
| Input | AirNet | PromptIR | **AdaIR** | Reference |

Figure 5: Image deraining results on Rain100L (Yang et al., 2019) between all-in-one methods. AdaIR yields high-fidelity rain-free images with structural fidelity and without streak artifacts.

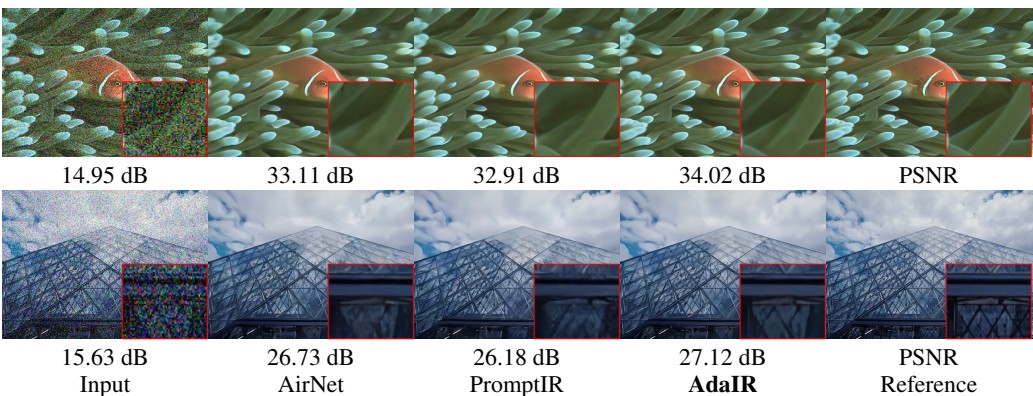

| 14.95 dB | 33.11 dB | 32.91 dB | 34.02 dB | PSNR |
| 15.63 dB | 26.73 dB | 26.18 dB | 27.12 dB | PSNR |
| Input | AirNet | PromptIR | **AdaIR** | Reference |

Figure 6: Image denoising comparisons on BSD68 (Martin et al., 2001) between all-in-one methods. The image reproduction quality of our AdaIR is more visually faithful to the ground truth.

Table 2: Comparisons for five-degradation all-in-one restoration. Denoising results are reported for the noise level $\sigma = 25$. The top super-row methods denote the general image restoration approaches, and the rest are specialized all-in-one approaches. On SOTS (Li et al., 2018) for dehazing, AdaIR attains a remarkable gain of 3.43 dB over InstructIR (Conde et al., 2024).

| Method | Dehazing on SOTS | | Deraining on Rain100L | | Denoising on BSD68 | | Deblurring on GoPro | | Low-Light on LOL | | Average | | |
| | PSNR | SSIM | PSNR | SSIM | PSNR | SSIM | PSNR | SSIM | PSNR | SSIM | PSNR | SSIM | Params |
| --- | --- | --- | --- | --- | --- | --- | --- | --- | --- | --- | --- | --- | --- |
| NAFNet (Chen et al., 2022) | 25.23 | 0.939 | 35.56 | 0.967 | 31.02 | 0.883 | 26.53 | 0.808 | 20.49 | 0.809 | 27.76 | 0.881 | 17.11M |
| HINet (Chen et al., 2021c) | 24.74 | 0.937 | 35.67 | 0.969 | 31.00 | 0.881 | 26.12 | 0.788 | 19.47 | 0.800 | 27.40 | 0.875 | - |
| MPRNet (Zamir et al., 2021) | 24.27 | 0.937 | 38.16 | 0.981 | 31.35 | 0.889 | 26.87 | 0.823 | 20.84 | 0.824 | 28.27 | 0.890 | 20.1M |
| DGUNet (Mou et al., 2022) | 24.78 | 0.940 | 36.62 | 0.971 | 31.10 | 0.883 | 27.25 | 0.837 | 21.87 | 0.823 | 28.32 | 0.891 | 17.33M |
| MIRNetV2 (Zamir et al., 2022b) | 24.03 | 0.927 | 33.89 | 0.954 | 30.97 | 0.881 | 26.30 | 0.799 | 21.52 | 0.815 | 27.34 | 0.875 | 5.86M |
| SwinIR (Liang et al., 2021) | 21.50 | 0.891 | 30.78 | 0.923 | 30.59 | 0.868 | 24.52 | 0.773 | 17.81 | 0.723 | 25.04 | 0.835 | 0.91M |
| Restormer (Zamir et al., 2022a) | 24.09 | 0.927 | 34.81 | 0.962 | 31.49 | 0.884 | 27.22 | 0.829 | 20.41 | 0.806 | 27.60 | 0.881 | 26.13M |
| DL (Fan et al., 2019) | 20.54 | 0.826 | 21.96 | 0.762 | 23.09 | 0.745 | 19.86 | 0.672 | 19.83 | 0.712 | 21.05 | 0.743 | 2.09M |
| Transweather (Valanarasu et al., 2022) | 21.32 | 0.885 | 29.43 | 0.905 | 29.00 | 0.841 | 25.12 | 0.757 | 21.21 | 0.792 | 25.22 | 0.836 | 37.93M |
| TAPE (Liu et al., 2022) | 22.16 | 0.861 | 29.67 | 0.904 | 30.18 | 0.855 | 24.47 | 0.763 | 18.97 | 0.621 | 25.09 | 0.801 | 1.07M |
| AirNet (Li et al., 2022) | 21.04 | 0.884 | 32.98 | 0.951 | 30.91 | 0.882 | 24.35 | 0.781 | 18.18 | 0.735 | 25.49 | 0.846 | 8.93M |
| IDR (Zhang et al., 2023) | 25.24 | 0.943 | 35.63 | 0.965 | 31.60 | 0.887 | 27.87 | 0.846 | 21.34 | 0.826 | 28.34 | 0.893 | 15.34M |
| PromptIR (Potlapalli et al., 2023) | 26.54 | 0.949 | 36.37 | 0.970 | 31.47 | 0.886 | 28.71 | 0.881 | 22.68 | 0.832 | 29.15 | 0.904 | 32.96M |
| Gridformer (Wang et al., 2024c) | 26.79 | 0.951 | 36.61 | 0.971 | 31.45 | 0.885 | 29.22 | 0.884 | 22.59 | 0.831 | 29.33 | 0.904 | 34.07M |
| InstructIR (Conde et al., 2024) | 27.10 | 0.956 | 36.84 | 0.973 | 31.40 | 0.887 | 29.40 | 0.886 | 23.00 | 0.836 | 29.55 | 0.907 | 15.80M |
| **AdaIR (Ours)** | 30.53 | 0.978 | 38.02 | 0.981 | 31.35 | 0.889 | 28.12 | 0.858 | 23.00 | 0.845 | 30.20 | 0.910 | 28.77M |

## 4.2 Additional All-in-One Results: Five Distinct Degradations

Following the recent work of IDR (Zhang et al., 2023), we further verify the effectiveness of AdaIR by performing experiments on five restoration tasks: dehazing, deraining, denoising, deblurring, and low-light image enhancement. For this, we train an all-in-one AdaIR model on combined datasets gathered for five different tasks. These include datasets from the aforementioned three-task setting as

Table 3: Image denoising results of directly applying the pre-trained model under the five-degradation setting to the Urban100 (Huang et al., 2015), Kodak24 (Rich, 1999) and BSD68 (Martin et al., 2001) datasets. The results are PSNR scores. On Urban100 (Huang et al., 2015) for the noise level $\sigma = 25$, AdaIR produces a significant performance gain of 0.39 dB PSNR over IDR (Zhang et al., 2023).

| Method | Urban100 $\sigma=15$ | $\sigma=25$ | $\sigma=50$ | Kodak24 $\sigma=15$ | $\sigma=25$ | $\sigma=50$ | BSD68 $\sigma=15$ | $\sigma=25$ | $\sigma=50$ | Average |
|---|---|---|---|---|---|---|---|---|---|---|
| DL (Fan et al., 2019) | 21.10 | 21.28 | 20.42 | 22.63 | 22.66 | 21.95 | 23.16 | 23.09 | 22.09 | 22.04 |
| TAPE (Liu et al., 2022) | 32.19 | 29.65 | 25.87 | 33.24 | 30.70 | 27.19 | 32.86 | 30.18 | 26.63 | 29.83 |
| AirNet (Li et al., 2022) | 33.16 | 30.83 | 27.45 | 34.14 | 31.74 | 28.59 | 33.49 | 30.91 | 27.66 | 30.89 |
| IDR (Zhang et al., 2023) | 33.82 | 31.29 | 28.07 | 34.78 | 32.42 | 29.13 | 34.11 | 31.60 | 28.14 | 31.48 |
| **AdaIR (Ours)** | 34.10 | 31.68 | 28.29 | 34.89 | 32.38 | 29.21 | 34.01 | 31.35 | 28.06 | 31.55 |

Table 4: Ablation studies for the proposed components. *Fixed* uses a fixed square mask with sides of 10. FLOPs are measured on the patch size of $256 \times 256 \times 3$.

| Net | FMiM Baseline | Fixed | MGB | FMoM L-H | H-L | PSNR | SSIM | Overhead Params. | FLOPs |
|---|---|---|---|---|---|---|---|---|---|
| (a) | ✓ | | | | | 28.21 | 0.966 | 26.13M | 141.24G |
| (b) | ✓ | ✓ | | | | 29.79 | 0.969 | 27.73M | 145.09G |
| (c) | ✓ | ✓ | | ✓ | | 30.37 | 0.975 | 28.74M | 147.44G |
| (d) | ✓ | ✓ | | ✓ | ✓ | 30.52 | 0.976 | 28.74M | 147.44G |
| (e) | ✓ | | ✓ | ✓ | ✓ | 31.24 | 0.978 | 28.77M | 147.45G |

Table 5: Spectra decomposition. *Adaptive* uses adaptive methods following (Zhou et al., 2024).

| Method | Pool | Gaussian | Adaptive | Ours |
|---|---|---|---|---|
| PSNR | 30.59 | 30.22 | 30.25 | 31.24 |

Table 6: Degradation sources.

| Method | Embedding | Ours |
|---|---|---|
| PSNR | 29.29 | 30.52 |
| SSIM | 0.969 | 0.976 |

well as additional datasets: GoPro (Nah et al., 2017) for motion deblurring, and LOL-v1 (Wei et al., 2018) for low-light image enhancement.

Table 2 shows that AdaIR achieves a 0.25 dB gain compared to the recent best method Instruc-tIR (Conde et al., 2024), when averaged across five restoration tasks. Particularly, the performance improvement is over 3 dB on dehazing. Table 3 reports denoising results on three different datasets with various noise levels. It can be seen that our method performs favorably well compared to the other competing approaches.

## 4.3 ABLATION STUDIES

In this section, we conduct ablation studies to test the impact of various individual components to the overall performance of AdaIR. All ablation experiments are performed on the image dehazing task by training models for 20 epochs.

**Impact of individual architecture modules.** Table 4 summarizes the performance benefits of individual architectural contributions. Table 4(b) demonstrates that the proposed frequency mining mechanism (FMiM) brings gains of 1.58 dB PSNR over the baseline model, using only a fixed mask to decompose the spectra of input images. Furthermore, the L-H unit boosts the performance to 30.37 dB PSNR; see Table 4(c). It can be seen in Table 4(d) that we use both L-H and H-L units, and the performance reaches 30.52 dB PSNR. Finally, Table 4(e) shows that the overall AdaIR brings 3.03 dB improvement over the baseline, while incurring a small computational overhead of 2.64M parameters and 6.21 GFlops. These results corroborate the effectiveness of our design.

**Strategies for spectral decomposition.** We carry out this ablation to test different strategies to segregate low- and high-frequency representations from the degraded input images. We compare the proposed mask-guided adaptive frequency decomposition approach with the Average pooling, Gaussian filtering, and Adaptive (Zhou et al., 2024) strategies. Results are provided in Table 5. Following (Cui et al., 2023a), we use average pooling to obtain the low-frequency features which are then subtracted from the input features to obtain the high-frequency features. This strategy provides a PSNR of 30.59 (see column 1 in Table 5), which is 0.65 dB lower than our method. Similarly, when we switch to the Gaussian filter of size $5 \times 5$, the model achieves only 30.22 dB PSNR (second column). Moreover, our method is superior to the alternative (Zhou et al., 2024) that uses dynamic spatial convolutions for spectral decomposition. Our method of applying a flexible mask for Fourier spectra decomposition performs the best, yielding 31.24 dB.

Table 7: Results on the unseen desnowing task with the CSD (Chen et al., 2021d) dataset.

| Method | AirNet | PromptIR | Ours |
|--------|--------|----------|------|
| PSNR | 19.32 | 20.47 | 20.54 |
| SSIM | 0.733 | 0.7638 | 0.7643 |

Table 8: Results of performing image denoising on the Rain100L dataset ($\sigma = 50$).

| Method | AirNet | PromptIR | Ours |
|--------|--------|----------|------|
| PSNR | 27.25 | 27.34 | 27.51 |
| SSIM | 0.790 | 0.791 | 0.798 |

| Blurry | 0.00016 | 0.00099 | 0.00257 | 0.00046 | 0.00561 | 0.00801 | 0.00234 | 0.00288 | 0.04374 |
|--------|---------|---------|---------|---------|---------|---------|---------|---------|---------|
| $\mathbf{A}_{H-L}$ | 0.72464 | 0.34558 | 0.89120 | 0.81372 | 0.80288 | 0.92845 | 0.59886 | 0.95265 | 0.93757 |

Figure 7: First column shows the blurry image and the spatial attention map in $\mathbf{A}_{H-L}$. Others are the channel-wise features before H-L and the corresponding attention scores in $\mathbf{A}_{L-H}$.

**Frequency representation mining at image-level vs. feature-level.** Each AFLB block in AdaIR decoder receives the original degraded image as input, on which FMiM applies the procedure of spectra decomposition. To verify the efficacy of this design, we switch to using the input embedding features $\mathbf{X}$ (rather than degraded image) for frequency representation. This ablation result in Table 6 shows a performance drop from $30.52$ dB to $29.29$ dB, indicating that the raw input image offers better discriminative information about the degradation for effective spectra separation.

**Generalization to out-of-distribution degradations.** To demonstrate the generalization ability of our AdaIR, we take the all-in-one model trained in the three-task setting and directly apply it to out-of-distribution datasets. Table 7 shows that, on the unseen task of image desnowing, AdaIR provides more favorable results than other approaches. We then perform image denoising using the Rain100L Yang et al. (2019) dataset instead of BSD68 Martin et al. (2001). Table 8 depicts that our method is more robust in the out-of-distribution scenes than PromptIR (Potlapalli et al., 2023) and AirNet (Li et al., 2022).

**Mechanism of FMoM.** In FMiM, we extract different frequency components from input features. These features are then categorized into low- and high-frequency groups using the dynamic, learnable module MGB, which adaptively adjusts the cutoff frequency boundary based on the specific degradation observed in the image. Once the low- and high-frequency features are segregated, they are processed by the FMoM. This module is responsible for either suppressing or allowing specific frequency components to pass through, depending on the nature of the degradation, effectively enhancing the restoration process. To better illustrate the interaction between frequency features, we visualize the attention weights generated by the High-to-Low (H-L) and Low-to-High (L-H) modules in Fig. 7. The high-frequency features, rich in spatial signals, assist the low-frequency branch in focusing on and effectively addressing severely impacted regions, such as the girl in the image. Conversely, the low-frequency features, which provide a global view, help the high-frequency features to avoid overemphasizing those challenging regions.

## 5 CONCLUSION

This paper introduces AdaIR, an all-in-one image restoration model capable of adaptively removing different kinds of image degradations. Motivated by the observation that different degradations affect distinct frequency bands, we have developed two novel components: a frequency mining module and a frequency modulation module. These modules are designed to identify and enhance the relevant frequency components based on the degradation patterns present in the input image. Specifically, the frequency mining module extracts specific frequency elements from the image's intermediate features, guided by an adaptive decomposition of the input's spectral characteristics that reflect the underlying degradation. Subsequently, the frequency modulation module further refines these elements by facilitating the exchange of complementary information across different frequency features. Incorporating the proposed modules into a U-shaped Transformer backbone, the proposed network achieves state-of-the-art performance on a range of image restoration tasks.

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

APPENDIX

This appendix provides generalization evaluation (Sec. A), more experimental results under the single-task setting (Sec. B), additional ablation studies (Sec. C), computational comparisons (Sec. D), visualization for FMiM (Sec. E), architectural details of the transformer block (Sec. F), and additional visual results (Sec. G).

More qualitative comparisons on different datasets are provided in the supplementary material.

## A  GENERALIZATION EVALUATION

We assess the generalization capability of our model on additional out-of-distribution degradations and compare the results against state-of-the-art all-in-one algorithms. As presented in Table 9, our method demonstrates superior performance on two previously unseen degradation types: defocus deblurring and raindrops. Additionally, we evaluate our approach on the real-world UAVDT (Du et al., 2018) dataset, which consists of images captured by UAVs at varying altitudes and exhibiting diverse levels of hazy degradation. Figure 8 illustrates that our model is more robust in real-world scenarios by restoring shaper images.

Table 9: Generalization evaluation of all-in-one algorithms. The models are trained under the three-task setting and directly applied to the DPDD (Abuolaim & Brown, 2020) and AGAN (Qian et al., 2018) datasets for defocus deblurring and raindrop removal, respectively.

| Method | DPDD (Abuolaim & Brown, 2020) | | AGAN (Qian et al., 2018) | |
| --- | --- | --- | --- | --- |
| | PSNR | SSIM | PSNR | SSIM |
| AirNet | 20.17 | 0.662 | 22.09 | 0.822 |
| PromptIR | 21.76 | 0.661 | 22.98 | 0.827 |
| Ours | 22.93 | 0.711 | 23.14 | 0.826 |

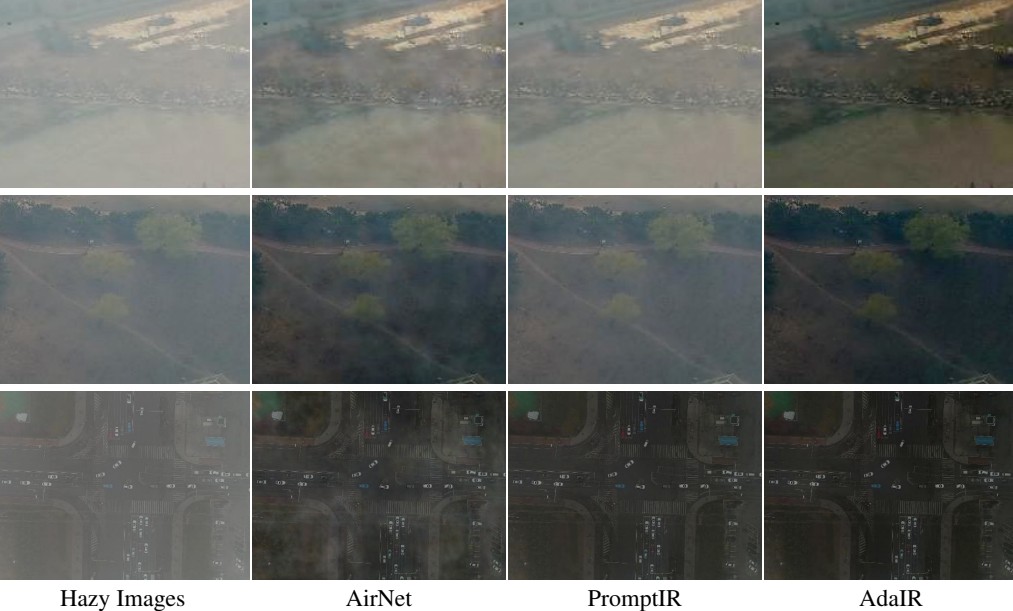

| Hazy Images | AirNet | PromptIR | AdaIR |

Figure 8: Visual comparisons on the UAVDT (Du et al., 2018) dataset.

## B  SINGLE DEGRADATION ONE-BY-ONE RESULTS

Consistent with previous works (Li et al., 2022; Potlapalli et al., 2023), we further evaluate AdaIR under the single-task experimental protocol. To this end, we train separate copies of the AdaIR model

for each restoration task. The numerical results on SOTS-Outdoor for image dehazing are presented in Table 10. Our method significantly outperforms previous state-of-the-art all-in-one approaches, PromptIR (Potlapalli et al., 2023) and AirNet (Li et al., 2022), by 0.49 dB and 8.62 dB, respectively, attributed to the adaptive frequency separation and modulation ability for haze degradations of different densities. Similarly, on the deraining task, Table 11 shows that our AdaIR advances the state-of-the-art (Potlapalli et al., 2023) by 1.86 dB. Compared to our baseline model (Zamir et al., 2022a), the accuracy gain is 2.16 dB PSNR, suggesting the efficacy of our designs. Furthermore, we provide experimental results for image denoising on two datasets with different noise levels. As can be seen in Table 12, our method yields an average performance gain of 0.13 dB PSNR over the strong competitor PromptIR. Compared to other methods, our method has more advantages on the Urban100 dataset than BSD68. This phenomenon is probably due to the higher resolution of Urban100 images, enabling more accurate frequency modulation.

Table 10: Dehazing results in the single-task setting on the SOTS-Outdoor (Li et al., 2018) dataset. Compared to PromptIR (Potlapalli et al., 2023), our method generates a 0.49 dB PSNR improvement.

| Method | DehazeNet | MSCNN | AODNet | EPDN | FDGAN | AirNet | Restormer | PromptIR | **AdaIR** |
|---|---|---|---|---|---|---|---|---|---|
| PSNR | 22.46 | 22.06 | 20.29 | 22.57 | 23.15 | 23.18 | 30.87 | 31.31 | 31.80 |
| SSIM | 0.851 | 0.908 | 0.877 | 0.863 | 0.921 | 0.900 | 0.969 | 0.973 | 0.981 |

Table 11: Deraining results in the single-task setting on the Rain100L (Yang et al., 2019) dataset. Our AdaIR obtains a significant performance boost of 1.86 dB PSNR over PromptIR (Potlapalli et al., 2023).

| Method | DIDMDN | UMR | SIRR | MSPFN | LPNet | AirNet | Restormer | PromptIR | **AdaIR** |
|---|---|---|---|---|---|---|---|---|---|
| PSNR | 23.79 | 32.39 | 32.37 | 33.50 | 33.61 | 34.90 | 36.74 | 37.04 | 38.90 |
| SSIM | 0.773 | 0.921 | 0.926 | 0.948 | 0.958 | 0.977 | 0.978 | 0.979 | 0.985 |

Table 12: Denoising results in the single-task setting on Urban100 (Huang et al., 2015) and BSD68 (Martin et al., 2001). On Urban100 (Huang et al., 2015) for the noise level 50, AdaIR yields a 0.31 dB gain over PromptIR (Potlapalli et al., 2023).

| Method | Urban100 | | | | | | BSD68 | | | | | | Average | |
|---|---|---|---|---|---|---|---|---|---|---|---|---|---|---|
| | $\sigma = 15$ | | $\sigma = 25$ | | $\sigma = 50$ | | $\sigma = 15$ | | $\sigma = 25$ | | $\sigma = 50$ | | | |
| | PSNR | SSIM | PSNR | SSIM | PSNR | SSIM | PSNR | SSIM | PSNR | SSIM | PSNR | SSIM | PSNR | SSIM |
| CBM3D (Dabov et al., 2007) | 33.93 | 0.941 | 31.36 | 0.909 | 27.93 | 0.840 | 33.50 | 0.922 | 30.69 | 0.868 | 27.36 | 0.763 | 30.80 | 0.874 |
| DnCNN (Zhang et al., 2017a) | 32.98 | 0.931 | 30.81 | 0.902 | 27.59 | 0.833 | 33.89 | 0.930 | 31.23 | 0.883 | 27.92 | 0.789 | 30.74 | 0.878 |
| IRCNN (Zhang et al., 2017b) | 27.59 | 0.833 | 31.20 | 0.909 | 27.70 | 0.840 | 33.87 | 0.929 | 31.18 | 0.882 | 27.88 | 0.790 | 29.90 | 0.864 |
| FFDNet (Zhang et al., 2018) | 33.83 | 0.942 | 31.40 | 0.912 | 28.05 | 0.848 | 33.87 | 0.929 | 31.21 | 0.882 | 27.96 | 0.789 | 31.05 | 0.884 |
| BRDNet (Tian et al., 2020) | 34.42 | 0.946 | 31.99 | 0.919 | 28.56 | 0.858 | 34.10 | 0.929 | 31.43 | 0.885 | 28.16 | 0.794 | 31.44 | 0.889 |
| AirNet (Li et al., 2022) | 34.40 | 0.949 | 32.10 | 0.924 | 28.88 | 0.871 | 34.14 | 0.936 | 31.48 | 0.893 | 28.23 | 0.806 | 31.54 | 0.897 |
| PromptIR (Potlapalli et al., 2023) | 34.77 | 0.952 | 32.49 | 0.929 | 29.39 | 0.881 | 34.34 | 0.938 | 31.71 | 0.897 | 28.49 | 0.813 | 31.87 | 0.902 |
| **AdaIR (Ours)** | 34.96 | 0.953 | 32.74 | 0.931 | 29.70 | 0.885 | 34.36 | 0.938 | 31.72 | 0.897 | 28.49 | 0.813 | 32.00 | 0.903 |

## C   ADDITIONAL ABLATION STUDIES

**AFLBs in encoder and decoder?** We run an experiment to assess the feasibility of employing AFLB modules on either the encoder side, decoder side, or both. Table 13 shows that utilizing AFLBs in both the encoder and decoder leads to notable performance degradation compared to AFLBs solely integrated into the decoder.

**Placement of AFLB in the network.** Next, we conduct an ablation experiment to study where to place AFLBs in our hierarchical network. Table 14 demonstrates that employing only one AFLB (between level 1 and level 2) leads to a deterioration in the network's performance (29.58 dB in the top row). Conversely, integrating AFLBs between every consecutive level of the decoder yields the best performance.

**Design choices of FMoM.** We investigate different choices for the frequency modulation module (FMoM). As shown in Fig. 9(a), we leverage the commonly used spatial attention (Woo et al., 2018)

Table 13: Comparisons of image dehazing under the single-task setting: between the use of AFLBs on either the encoder-side, decoder-side, or both.

| Method | Dehazing on SOTS (Li et al., 2018) | |
| --- | --- | --- |
| | PSNR | SSIM |
| Encoder+Decoder+AFLB | 29.70 | 0.973 |
| AdaIR (Ours) | 30.52 | 0.976 |

Table 14: AFLB position. Results are reported on the SOTS (Li et al., 2018) dataset.

| Method | PSNR | SSIM |
| --- | --- | --- |
| Level 2 | 28.58 | 0.973 |
| Level 2+3 | 29.83 | 0.975 |
| Level 2+3+4 | 30.52 | 0.976 |

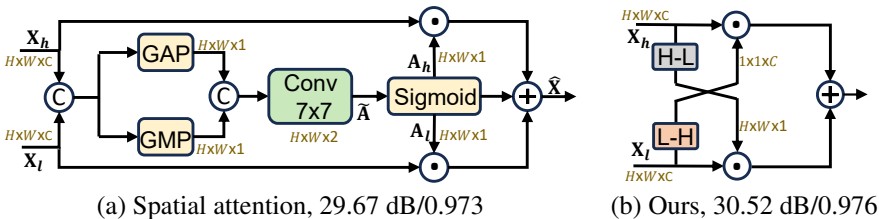

(a) Spatial attention, 29.67 dB/0.973      (b) Ours, 30.52 dB/0.976

Figure 9: Different choices for FMoM. (a) Using widely adopted spatial attention (Woo et al., 2018) to modulate different frequency features, where the attention map is generated without discriminating different frequency inputs. (b) Using specially designed attention units to exchange complementary information across different frequency features. GAP and GMP denote the global average pooling and global max pooling, respectively. The experiments are conducted on image dehazing under the single-task setting.

to modulate different frequency features without discriminating different inputs. Overall, the process is formally given by:

$$\hat{\mathbf{X}} = \mathbf{X}_h \odot \mathbf{A}_h + \mathbf{X}_l \odot \mathbf{A}_l, \qquad \text{where,} \tag{9}$$

$$\mathbf{A}_h, \mathbf{A}_l = \text{Split}\left(\delta(\widetilde{\mathbf{A}})\right), \qquad \text{where,} \tag{10}$$

$$\widetilde{\mathbf{A}} = W^{7 \times 7}\left([\text{GAP}([\mathbf{X}_h, \mathbf{X}_l]), \text{GMP}([\mathbf{X}_h, \mathbf{X}_l])]\right) \tag{11}$$

where $\odot$ represents element-wise multiplication, Split indicates splitting the features among the channel dimension, $\delta$ is the Sigmoid function, $W^{7 \times 7}$ is a $7 \times 7$ convolution, and $[\cdot, \cdot]$ is a concatenation operator. GAP and GMP are global average pooling and global max pooling among the channel dimensions, respectively. The experiments are performed on the image dehazing task under the single-task setting. This variant achieves only 29.67 dB PSNR, which is 0.85 dB lower than our FMoM, shown in Fig. 9(b), indicating the effectiveness of our design.

Furthermore, we conducted experiments to evaluate the impact of using different attention strategies in the two branches. As shown in Table 15, employing the same attention mechanism in both branches results in lower performance compared to our approach. This highlights the effectiveness of performing frequency interactions tailored to the distinct properties of different frequency components.

**Combinations of different degradations.** We investigate the influence of various combinations of degradation types on model performance, as presented in Table 16. As expected, including more degradation types make it increasingly difficult for the model to perform restoration. Notable, hazy images in a combined dataset lead to a larger performance drop than rainy or noisy images. One reason could be that the aim of the restoration model in deraining and denoising tasks is to focus

Table 15: Comparisons between different attention types.

| Unit | Attention Type | PSNR |
|---|---|---|
| (a) H-L/L-H | Channel/Channel | 30.10 |
| (b) H-L/L-H | Spatial/Spatial | 30.36 |
| (c) H-L/L-H (Ours) | Spatial/Channel | 30.52 |

more on restoring high-frequency content (noise, rain), whereas, in the dehazing task the goal is to focus on removing low-frequency (hazy) content.

Table 16: Ablation studies on the combinations of degradations for the three-task setting. Results are presented in the form of PSNR (dB)/SSIM.

| Degradation | | | Denoising on BSD68 | | | Deraining on | Dehazing |
|---|---|---|---|---|---|---|---|
| Noise | Rain | Haze | $\sigma = 15$ | $\sigma = 25$ | $\sigma = 50$ | on Rain100L | on SOTS |
| ✓ | | | 34.36/0.938 | 31.72/0.897 | 28.49/0.813 | - | - |
| | ✓ | | - | - | - | 38.90/0.985 | - |
| | | ✓ | - | - | - | - | 31.80/0.981 |
| ✓ | ✓ | | 34.31/0.938 | 31.67/0.896 | 28.42/0.811 | 38.22/0.983 | - |
| ✓ | | ✓ | 34.11/0.935 | 31.48/0.892 | 28.19/0.802 | - | 30.89/0.980 |
| | ✓ | ✓ | - | - | - | 38.44/0.983 | 30.54/0.978 |
| ✓ | ✓ | ✓ | 34.12/0.935 | 31.45/0.892 | 28.19/0.802 | 38.64/0.983 | 31.06/0.980 |

## D  COMPUTATIONAL COMPARISONS

Table 17 shows that the proposed AdaIR strikes a better tradeoff between accuracy and complexity than other all-in-one competing methods.

Table 17: Computational comparisons of all-in-one methods under the three-degradation setting. The average PSNR across three tasks is reported here (see Table 1 of the main paper for more detailed results). FLOPs are measured on the patch size of $256 \times 256 \times 3$.

| Method | Params. (M) | FLOPs (G) | PSNR |
|---|---|---|---|
| AirNet (Li et al., 2022) | 8.93 | 311 | 31.20 |
| PromptIR (Potlapalli et al., 2023) | 35.59 | 158.4 | 32.06 |
| AdaIR | 28.77 | 147.45 | 32.69 |

## E  VISUALIZATION FOR FMIM

Figure 10 visualizes the FMiM process, illustrating how various frequency components are separated from the input image and extracted from the features. Specifically, MGB produces a mask to decouple the input image into different frequencies (②, ③). Next, the obtained spectra are used to extract corresponding features from the input features (④), as shown in ⑤ and ⑥, which then interact in FMoM. The visualizations demonstrate the efficacy of our design. Additional examples of frequency decomposition for low-light image enhancement and dehazing tasks are provided in Figure 11. As shown, our model adaptively decouples images into different frequency bands. Furthermore, Figure 12 illustrates comparisons of features obtained before and after our AFLB module. The results demonstrate that our module effectively generates sharper features, contributing to high-fidelity reconstruction.

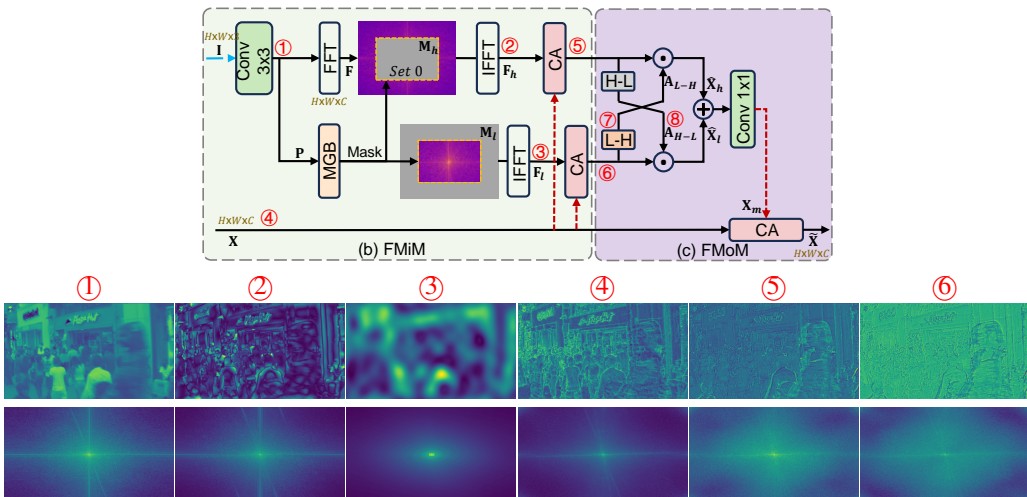

Figure 10: Visualizations for intermediate features and spectra. Our modules can decouple the image/features into different frequencies as expected. Attention weights in ⑦ and ⑧ are shown in Fig. 7 of the main paper.

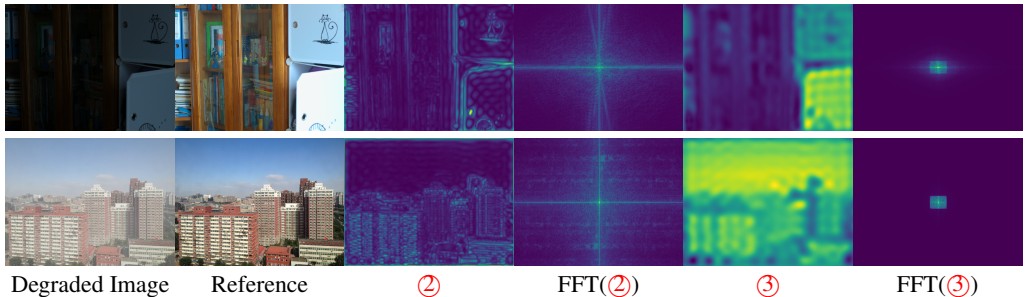

Figure 11: Visualizations of frequency decoupling. The two images are obtained from the LOL-v1 (Wei et al., 2018) and SOTS (Li et al., 2018) datasets for low-light image enhancement and dehazing, respectively.

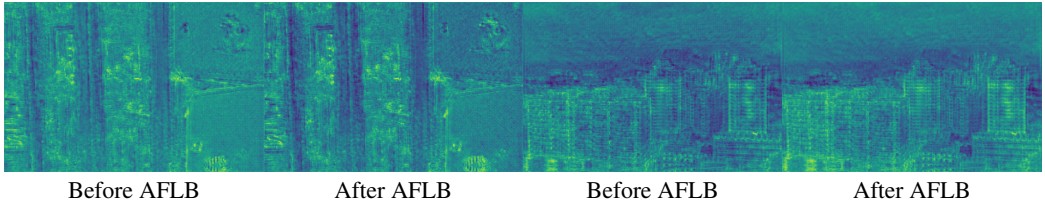

Figure 12: Feature comparisons based on the two images in Figure 11. Our module generates sharper features.

## F TRANSFORMER BLOCK IN THE ADAIR FRAMEWORK

In the AdaIR framework, we use Transformer Blocks (TB) based on the design proposed in (Zamir et al., 2022a). Fig. 13 presents its architectural details. It consists of two successive components, multi-dconv head transposed attention (MDTA) and gated-dconv feed-forward network (GDFN).

MDTA first normalizes the input $\mathbf{X} \in \mathbb{R}^{H \times W \times C}$ using a layer normalization operator (Ba et al., 2016), and then generates the *query* ($\mathbf{Q} \in \mathbb{R}^{H \times W \times C}$), *key* ($\mathbf{K} \in \mathbb{R}^{H \times W \times C}$), and *value* ($\mathbf{V} \in \mathbb{R}^{H \times W \times C}$) projections using combinations of $1 \times 1$ convolution and $3 \times 3$ depth-wise convolution layers. The transposed-attention map of size $C \times C$ is yielded by applying the Softmax function to the dot-product

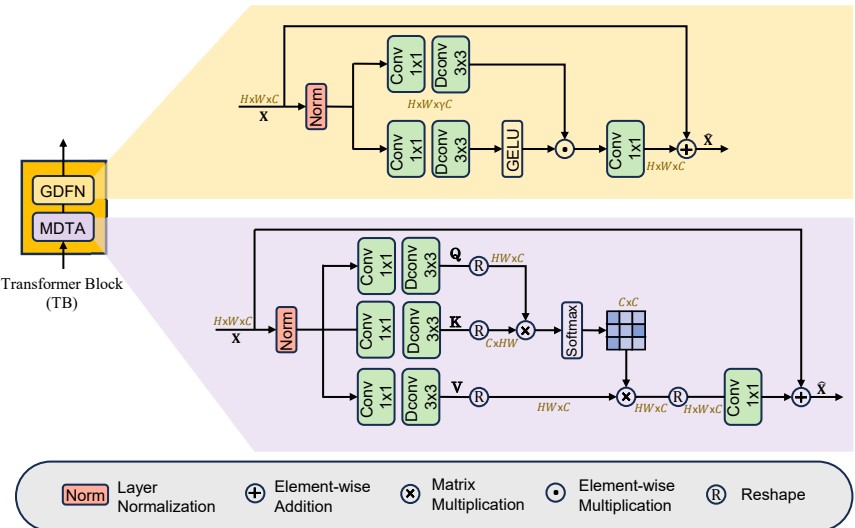

Figure 13: Architectural details of the Transformer Block (TB) (Zamir et al., 2022a) used in the AdaIR framework. TB involves two elements: multi-dconv head transposed attention (MDTA) and gated-dconv feed-forward network (GDFN).

results of the reshaped query and key projections. Overall, the process of MDTA is given by:

$$\hat{\mathbf{X}} = W_1^{1 \times 1} \text{Attention}\left(\mathbf{Q}', \mathbf{K}', \mathbf{V}'\right) + \mathbf{X}, \qquad \text{where,} \tag{12}$$

$$\text{Attention}\left(\mathbf{Q}', \mathbf{K}', \mathbf{V}'\right) = \mathbf{V}' \cdot \text{Softmax}\left(\mathbf{K}' \cdot \mathbf{Q}'/\alpha\right), \tag{13}$$

where $\hat{\mathbf{X}}$ is the output of MDTA. $W_1^{1 \times 1}$ denotes a $1 \times 1$ convolution. $\alpha$ is a learnable factor to control the magnitude of the dot product result of $\mathbf{K}$ and $\mathbf{Q}$. $\mathbf{Q}'$, $\mathbf{K}'$ and $\mathbf{V}'$ are obtained by reshaping tensors from the original size $\mathbb{R}^{H \times W \times C}$.

Similarly, GDFN first applies a layer normalization operator to normalize the input $\mathbf{X} \in \mathbb{R}^{H \times W \times C}$. The result then passes through two branches, each including a $1 \times 1$ convolution with a factor $\gamma$ to expand channels, followed by a $3 \times 3$ depth-wise convolution layer. Two branches converge using element-wise multiplication after activating one branch via a GELU function. Overall, the GDFN process is formally expressed as:

$$\hat{\mathbf{X}} = W_2^{1 \times 1} \text{Gating}(\mathbf{X}) + \mathbf{X}, \qquad \text{where,} \tag{14}$$

$$\text{Gating}(\mathbf{X}) = \phi\left(DW_1^{3 \times 3}\left(W_3^{1 \times 1}(\text{LN}(\mathbf{X}))\right)\right) \odot DW_2^{3 \times 3}\left(W_4^{1 \times 1}(\text{LN}(\mathbf{X}))\right), \tag{15}$$

where LN is the layer normalization, $\odot$ denotes element-wise multiplication, $DW^{3 \times 3}$ represents a $3 \times 3$ depth-wise convolution, and $\phi$ indicates the GELU non-linearity.

## G  ADDITIONAL VISUAL RESULTS

In this section, we provide the t-SNE result of our method under the five-degradation setting in Fig. 14. It can be seen that our method is capable of discriminating degradation contexts for five different degradation types. It is worth noting that the cluster for low-light image enhancement is closer to the dehazing cluster than others, suggesting the effectiveness of our model, since these two degradation types mainly impact the image content on low-frequency components.

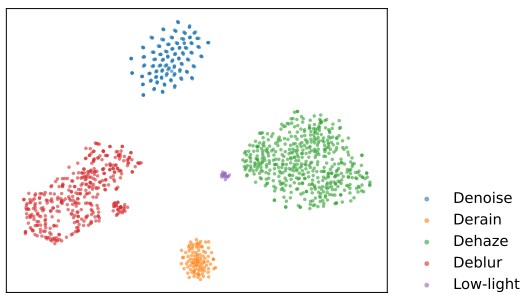

Figure 14: The t-SNE result of our model under the five-degradation setting.

