# AdaIR: Adaptive All-in-One Image Restoration via Frequency Mining and Modulation —Supplementary Material

**Yuning Cui**[1], **Syed Waqas Zamir**[2], **Salman Khan**[3,4],
**Alois Knoll**[1], **Mubarak Shah**[5], **Fahad Shahbaz Khan**[3,6]
[1]Technical University of Munich          [2]Inception Institute of Artificial Intelligence
[3]Mohammed Bin Zayed University of AI    [4]Australian National University
[5]University of Central Florida          [6]Linköping University

In this supplementary material, we provide more qualitative results of the all-in-one setting and single-task setting for three image restoration tasks, including image deraining, dehazing, and denoising.

## REFERENCES

Boyi Li, Wenqi Ren, Dengpan Fu, Dacheng Tao, Dan Feng, Wenjun Zeng, and Zhangyang Wang. Benchmarking single-image dehazing and beyond. *TIP*, 2018.

Boyun Li, Xiao Liu, Peng Hu, Zhongqin Wu, Jiancheng Lv, and Xi Peng. All-in-one image restoration for unknown corruption. In *CVPR*, 2022.

David Martin, Charless Fowlkes, Doron Tal, and Jitendra Malik. A database of human segmented natural images and its application to evaluating segmentation algorithms and measuring ecological statistics. In *ICCV*, 2001.

Wenhan Yang, Robby T Tan, Jiashi Feng, Zongming Guo, Shuicheng Yan, and Jiaying Liu. Joint rain detection and removal from a single image with contextualized deep networks. *TPAMI*, 2019.

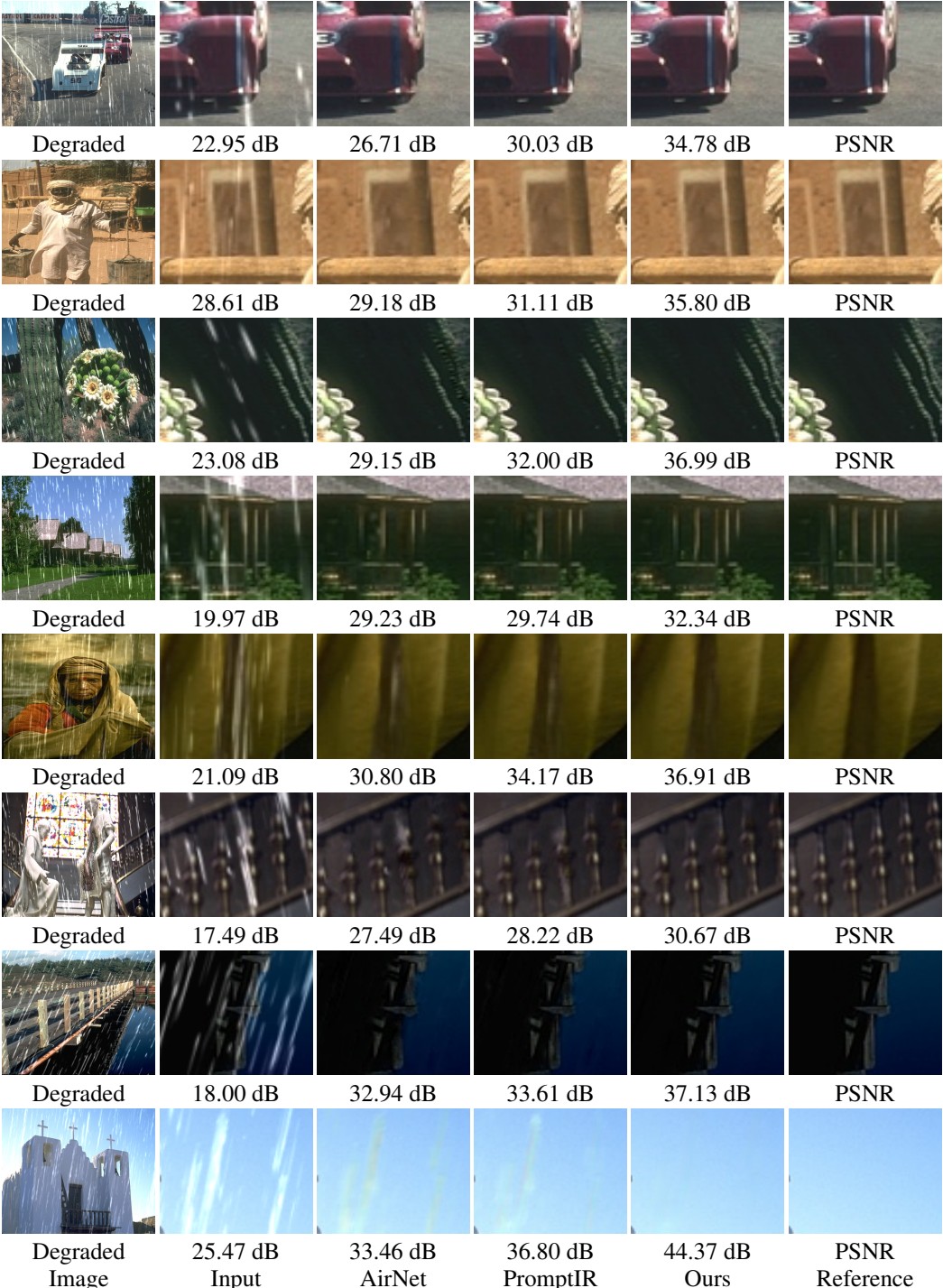

| | | | | | |
|---|---|---|---|---|---|
| Degraded | 22.95 dB | 26.71 dB | 30.03 dB | 34.78 dB | PSNR |
| Degraded | 28.61 dB | 29.18 dB | 31.11 dB | 35.80 dB | PSNR |
| Degraded | 23.08 dB | 29.15 dB | 32.00 dB | 36.99 dB | PSNR |
| Degraded | 19.97 dB | 29.23 dB | 29.74 dB | 32.34 dB | PSNR |
| Degraded | 21.09 dB | 30.80 dB | 34.17 dB | 36.91 dB | PSNR |
| Degraded | 17.49 dB | 27.49 dB | 28.22 dB | 30.67 dB | PSNR |
| Degraded | 18.00 dB | 32.94 dB | 33.61 dB | 37.13 dB | PSNR |
| Degraded Image | 25.47 dB Input | 33.46 dB AirNet | 36.80 dB PromptIR | 44.37 dB Ours | PSNR Reference |

Figure 1: Image deraining comparisons on Rain100L (Yang et al., 2019) under the three-degradation setting.

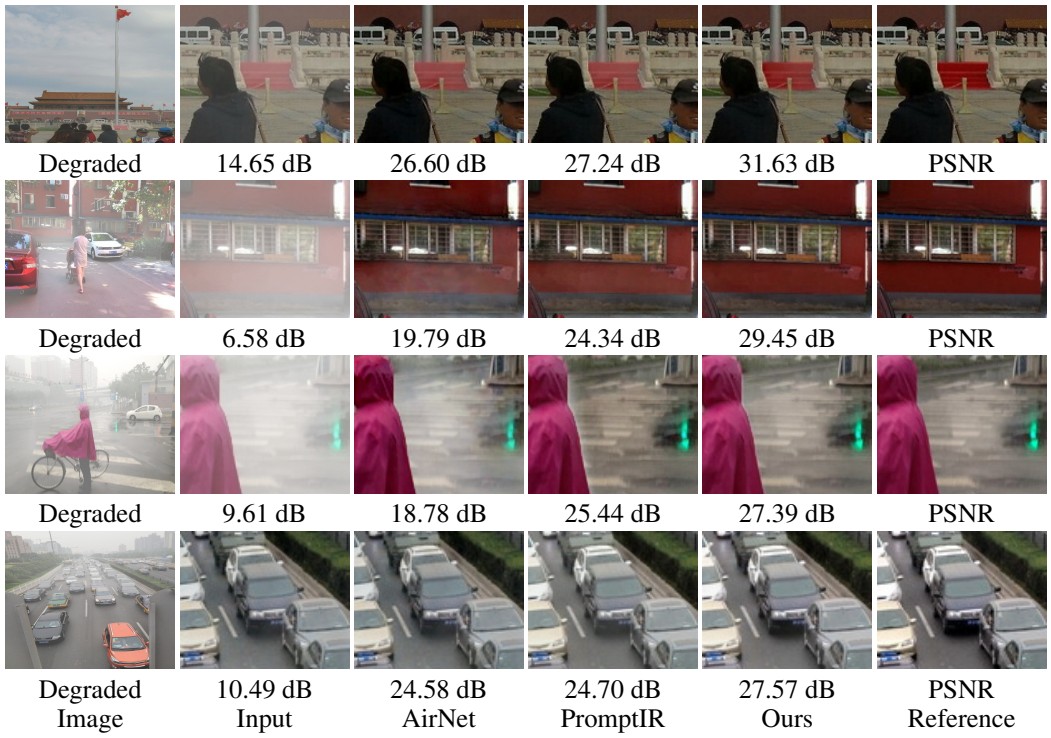

| Degraded | 14.65 dB | 26.60 dB | 27.24 dB | 31.63 dB | PSNR |
| Degraded | 6.58 dB | 19.79 dB | 24.34 dB | 29.45 dB | PSNR |
| Degraded | 9.61 dB | 18.78 dB | 25.44 dB | 27.39 dB | PSNR |
| Degraded Image | 10.49 dB Input | 24.58 dB AirNet | 24.70 dB PromptIR | 27.57 dB Ours | PSNR Reference |

Figure 2: Image dehazing comparisons on SOTS (Li et al., 2018) under the three-degradation setting.

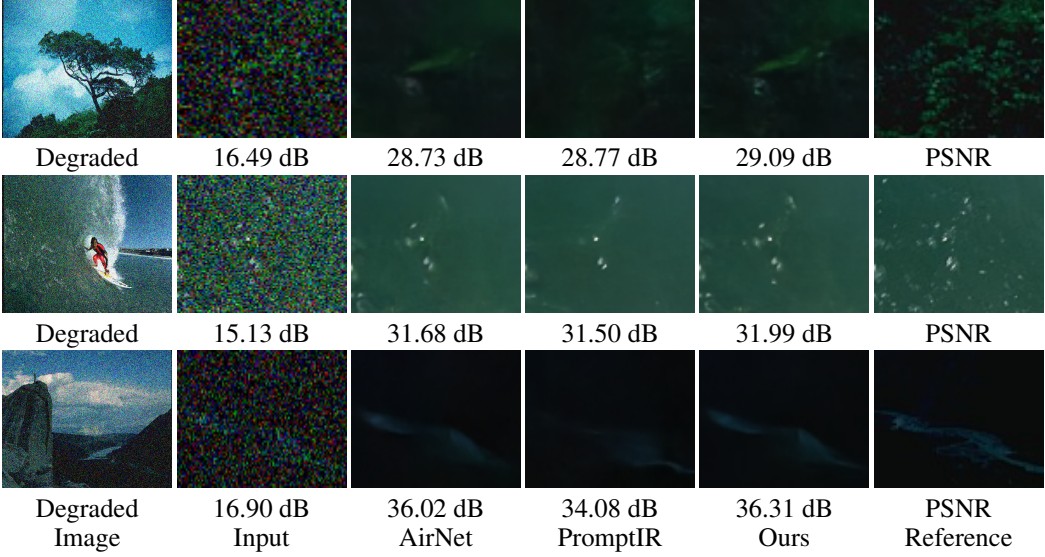

| Degraded | 16.49 dB | 28.73 dB | 28.77 dB | 29.09 dB | PSNR |
| Degraded | 15.13 dB | 31.68 dB | 31.50 dB | 31.99 dB | PSNR |
| Degraded Image | 16.90 dB Input | 36.02 dB AirNet | 34.08 dB PromptIR | 36.31 dB Ours | PSNR Reference |

Figure 3: Image denoising comparisons on BSD68 (Martin et al., 2001) with $\sigma = 50$ under the three-degradation setting.

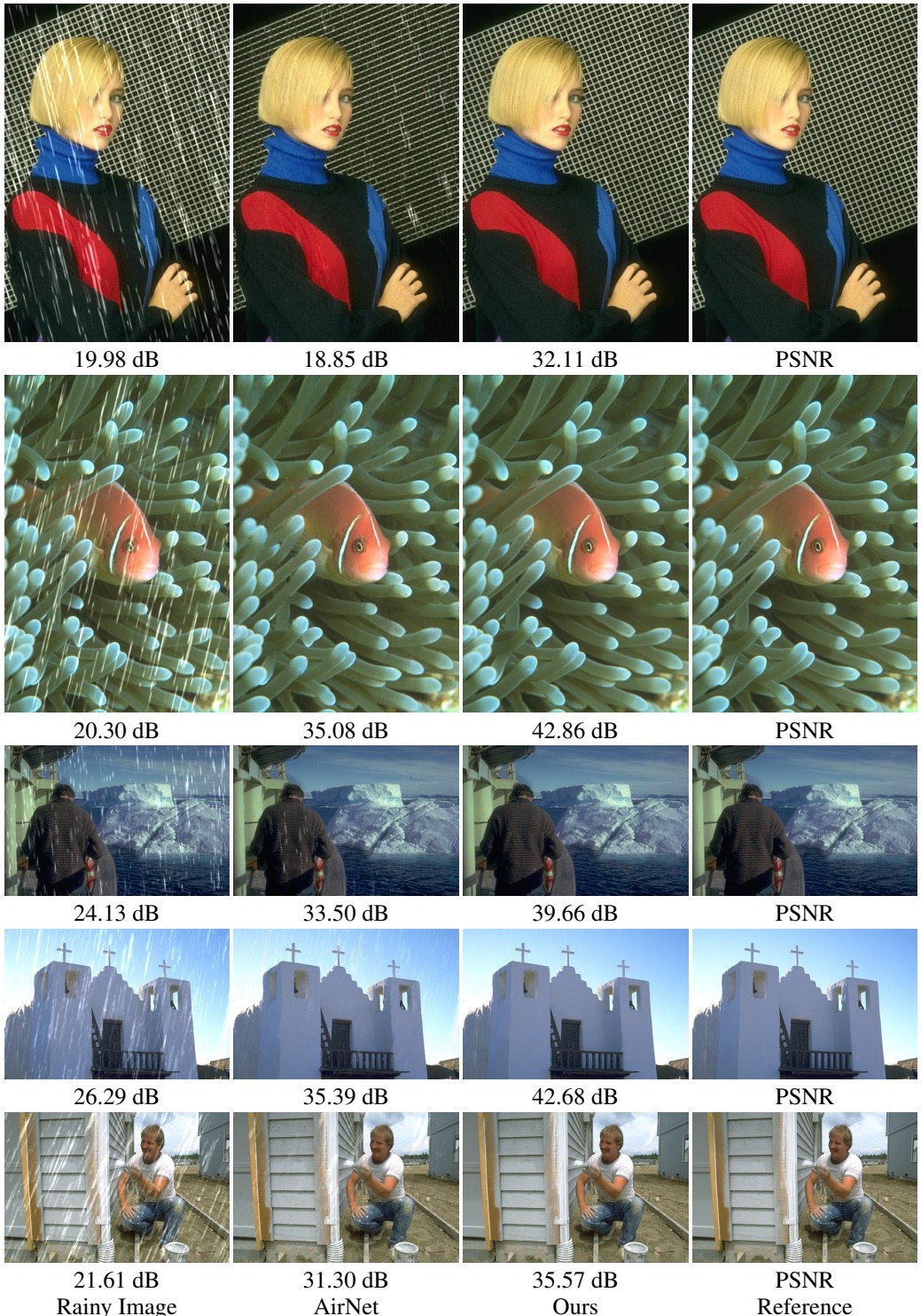

| 19.98 dB | 18.85 dB | 32.11 dB | PSNR |
| 20.30 dB | 35.08 dB | 42.86 dB | PSNR |
| 24.13 dB | 33.50 dB | 39.66 dB | PSNR |
| 26.29 dB | 35.39 dB | 42.68 dB | PSNR |
| 21.61 dB | 31.30 dB | 35.57 dB | PSNR |
| Rainy Image | AirNet | Ours | Reference |

Figure 4: Image draining comparisons under the single task setting on Rain100L (Yang et al., 2019).

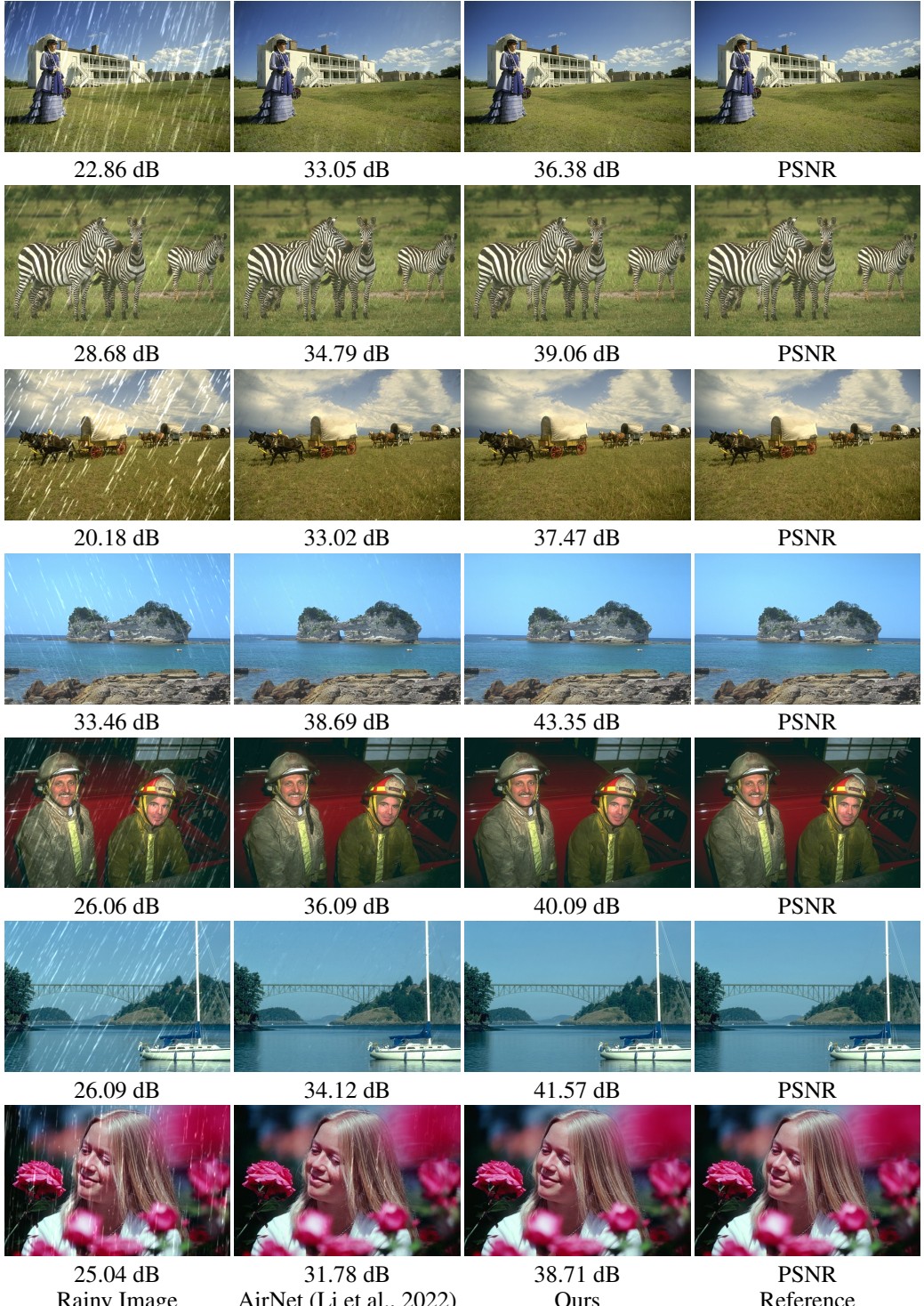

| | | | |
|---|---|---|---|
| 22.86 dB | 33.05 dB | 36.38 dB | PSNR |
| 28.68 dB | 34.79 dB | 39.06 dB | PSNR |
| 20.18 dB | 33.02 dB | 37.47 dB | PSNR |
| 33.46 dB | 38.69 dB | 43.35 dB | PSNR |
| 26.06 dB | 36.09 dB | 40.09 dB | PSNR |
| 26.09 dB | 34.12 dB | 41.57 dB | PSNR |
| 25.04 dB | 31.78 dB | 38.71 dB | PSNR |
| Rainy Image | AirNet (Li et al., 2022) | Ours | Reference |

Figure 5: Image draining comparisons under the single task setting on Rain100L (Yang et al., 2019).

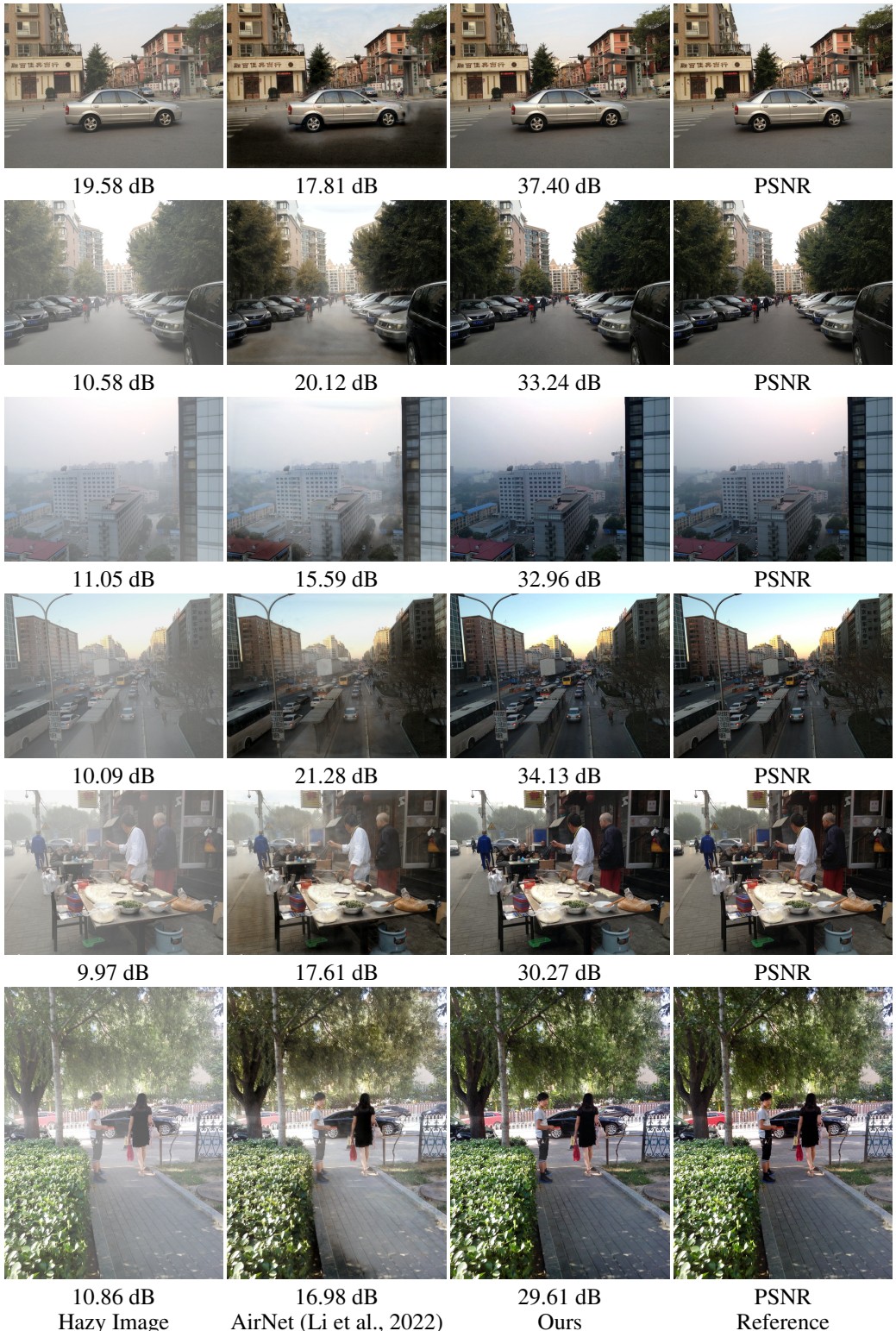

Figure 6: Image dehazing comparisons under the single task setting on SOTS (Li et al., 2018).

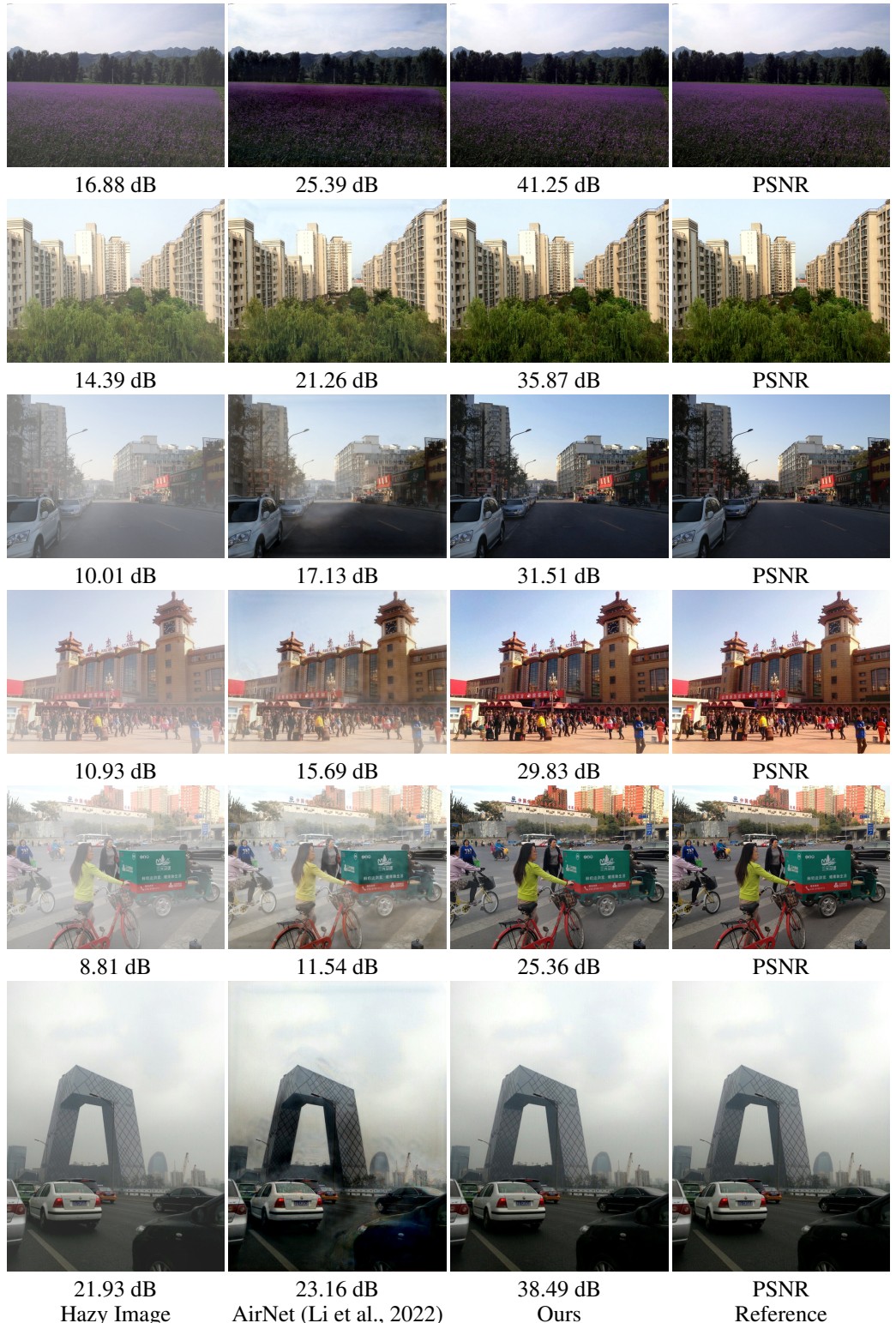

| 16.88 dB | 25.39 dB | 41.25 dB | PSNR |
| 14.39 dB | 21.26 dB | 35.87 dB | PSNR |
| 10.01 dB | 17.13 dB | 31.51 dB | PSNR |
| 10.93 dB | 15.69 dB | 29.83 dB | PSNR |
| 8.81 dB | 11.54 dB | 25.36 dB | PSNR |
| 21.93 dB | 23.16 dB | 38.49 dB | PSNR |
| Hazy Image | AirNet (Li et al., 2022) | Ours | Reference |

Figure 7: Image dehazing comparisons under the single task setting on SOTS (Li et al., 2018).

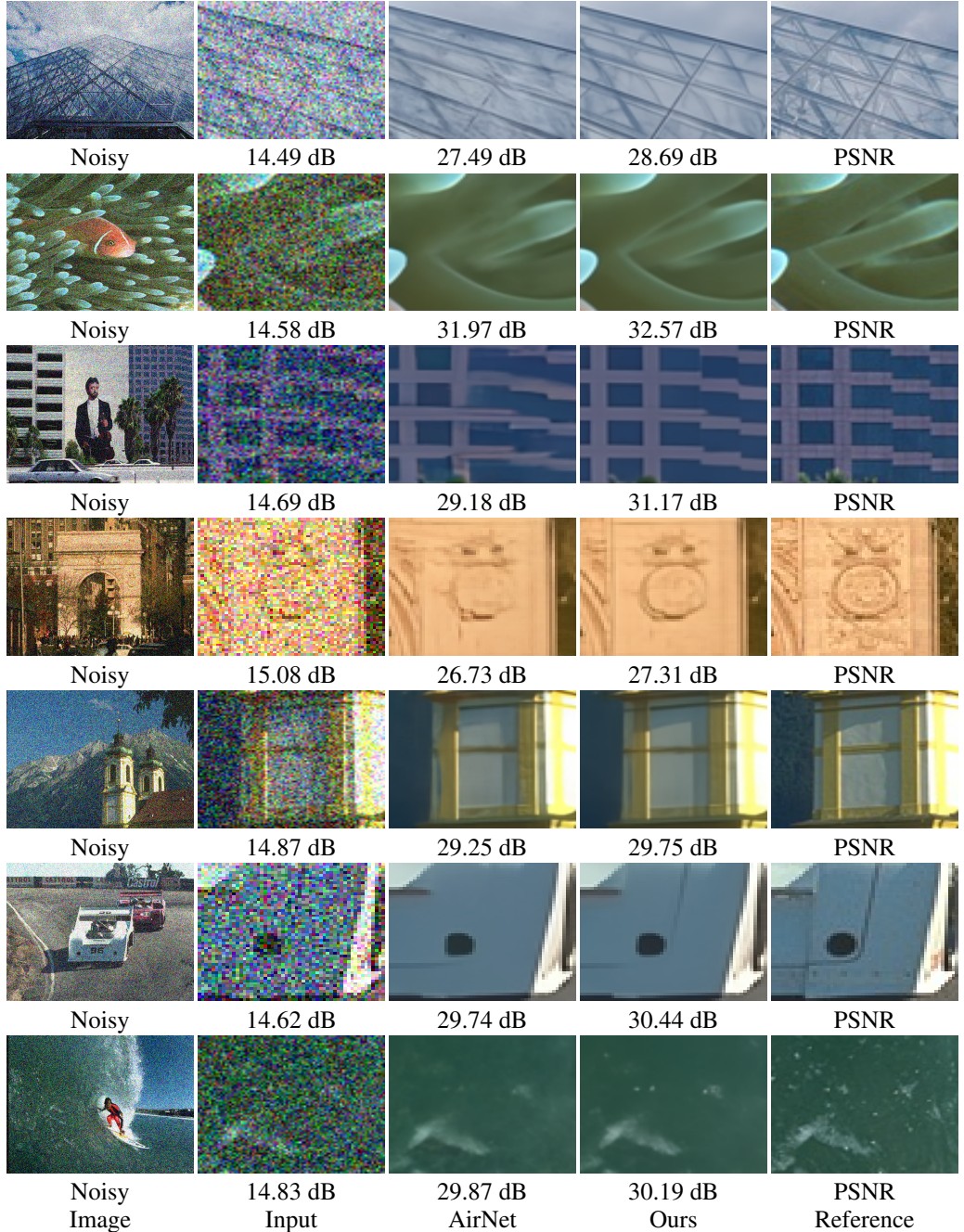

Figure 8: Image denoising results under single task setting on BSD68 (Martin et al., 2001) with $\sigma = 50$.