# OpenReview forum: "AdaIR: Adaptive All-in-One Image Restoration via Frequency Mining and Modulation"
_ICLR.cc/2025/Conference — ICLR 2025 Poster_

### Official Review · Reviewer_AFeH · 2024-10-29

**Soundness:** 2
**Presentation:** 2
**Contribution:** 2
**Rating:** 5
**Confidence:** 5

**Summary:**

This work solves the all-in-one problem in image restoration from a frequency perspective and demonstrates its effectiveness. But my main problem is that I think the insights and designed modules are fallen behind and there are already many homogenized work. Therefore, the novelty of this paper does not appeal to me.

**Strengths:**

The AdaIR model offers significant advantages by adaptively addressing various types of image degradations through innovative frequency analysis. Its frequency mining module precisely extracts relevant frequency components that reflect underlying degradations, while the frequency modulation module optimizes these components by allowing the exchange of complementary information across different features. Integrated into a U-shaped Transformer backbone, these components enable AdaIR to achieve state-of-the-art performance across diverse image restoration tasks, highlighting its robust adaptability and effectiveness.

**Weaknesses:**

1. In the t-sne clustering in Figure 2, the model is better at learning discriminative degradation contexts. First, the comparison method is relatively old, so some more recent comparison methods is suggested in the t-SNE analysis.

2. Moreover, I do not think that this completely separated state can help solve the all-in-one problem more effectively. In fact, the correlation between various degradations may need to be learned to solve it better. For example, an image of rainy scene should also have heavy haze, so these two degradations should have something in common. And the rain lines can be regarded as a kind of noise, so they may also be correlated in AirNet and PromptIR of Fig 2, rather than completely decoupled.  In other word, I doubt that the reason for the decoupling of the t-sne analysis may be the overfitting of the model on each dataset.

3. As the core motivation of this work, the idea in Figure 1 has been discussed extensively, and I think it is not innovative. Please see the question section for detailed discussion.

4. The method design is also based on the combination of the very traditional U-Net network and the attention mechanism. This design has obviously fallen behind. Diffusion model may be a more proper framework to solve this problem.

**Questions:**

Some works as following also have discussed using the frequency domain perspective to solve various degradation problems. [1] and [2] observe a specific degradation from the perspective of frequency domain.  [3] and [4] also adopt frequency mining and modulation to solve various degradations condition. It is suggested  to provide a point-by-point comparison of the proposed method with these works, highlighting key similarities and differences in approach and performance.

[1] Efficient Frequency Domain-Based Transformers for High-Quality Image Deblurring, CVPR, 2023
[2] Wavelet-based Fourier Information Interaction with Frequency Diffusion Adjustment for Underwater Image Restoration, CVPR, 2024
[3] Hybrid Frequency Modulation Network for Image Restoration, IJCAI, 2024
[4] Selective Frequency Network for Image Restoration, ICLR, 2023

---

> ### Author Response · Authors · 2024-11-28
> **Response to Reviewer AFeH**
>
> ``R4.1``:*t-SNE results*:
>
> Following the suggestion, we have incorporated a new method [5] for t-SNE comparisons, as shown in Figure 2 of the paper. Our method is better at learning discriminative degradation contexts.
>
> ``R4.2``:*Rationale of t-SNE*:
>
> The use of t-SNE is well-established in all-in-one algorithms [6,7] as a means of evaluating the ability to distinguish between different degradation types. While we acknowledge the existence of correlations between degradations, we believe this does not necessarily conflict with the goal of decoupling these degradations. Instead, the goal is to strike a balance; capturing these correlations while maintaining sufficient separation for effective restoration.
>
> As shown in Figure 14, the t-SNE results for the five-task setting demonstrate this balance. The correlations among degradations are reflected in the proximity of clusters. For instance, the cluster for low-light image enhancement is closer to the dehazing cluster, as both degradations predominantly affect the low-frequency components of the image. This proximity effectively captures the underlying relationship between these tasks while ensuring that individual degradations remain distinguishable.
>
> ``R4.3``:*Why UNet?*:
>
> Several recent all-in-one works (Art$_{\text{PromptIR}}$, PromptIR, TransWeather, DL, InstructIR) employ a hierarchical architecture. In our AdaIR, we also use UNet design and demonstrate how to simply add AFLB into the main network. This can easily be extended to other networks as well.
>
> Regarding the use of diffusion models versus regression-based methods, the choice depends on the intent of image reconstruction. If the primary goal is to preserve the original content without introducing artifacts or hallucinations (as is crucial in image restoration tasks), a regression-based method like AdaIR is more suitable. Diffusion models, while powerful, are generative in nature and prone to hallucinating information, which poses challenges in scenarios where fidelity/faithfulness to the original content is critical.
>
> ``R4.4``:*Differences with existing algorithms*:
>
> Ref.[1] introduces a Transformer-based architecture that leverages spectral multiplication for image deblurring. However, this method lacks mechanisms for interaction across different frequency bands, and its frequency quantization matrix remains static after training, limiting the model's adaptability to varying input characteristics. In contrast, Ref.[2] proposes a wavelet-based model for underwater image restoration, which explores frequency properties through frequency component swapping in the frequency domain. Despite its focus on frequency manipulation, the objectives and methodologies of Ref.[2] differ substantially from ours. Similarly, Ref.[3] employs global average pooling for feature separation but is not designed for all-in-one restoration scenarios. Its binary separation strategy is constrained to the lowest frequency and its complement, resulting in limited adaptability to diverse degradation types. Ref.[4] presents a CNN-based algorithm for general image restoration, utilizing adaptive convolution and spatial softmax operations. However, it does not comprehensively address the representation and interaction of a broad spectrum of frequency bands. When we replaced our frequency separation approach with the method from Ref.[4], the performance decreased by 0.99 dB in PSNR, further highlighting its limitations. In contrast, our approach performs frequency separation by applying adaptively generated masks to Fourier features, enabling comprehensive coverage across a wide range of frequency bands and ensuring adaptability to diverse degradation types. Additionally, we integrate tailored attention mechanisms to enhance frequency integration, effectively harnessing the unique properties of individual frequency components.
>
> [1] Kong et al., Efficient Frequency Domain-Based Transformers for High-Quality Image Deblurring, CVPR, 2023.
>
> [2] Zhao et al., Wavelet-based Fourier Information Interaction with Frequency Diffusion Adjustment for Underwater Image Restoration, CVPR 2024.
>
> [3] Cui et al., Hybrid Frequency Modulation Network for Image Restoration, IJCAI 2024.
>
> [4] Cui et al., Selective Frequency Network for Image Restoration, ICLR 2023.
>
> [5] Xu et al., Unified-width adaptive dynamic network for all-in-one image restoration, arXiv 2024.
>
> [6] Potlapalli et al., Promptir: Prompting for all-in-one image restoration, NeurIPS 2023.
>
> [7] Ai et al., Multimodal Prompt Perceiver: Empower Adaptiveness Generalizability and Fidelity for All-in-One Image Restoration, CVPR 2024.

---

> ### Author Response · Authors · 2024-12-02
> **We thank the reviewer again for the valuable feedback and happy to address any remaining concerns.**
>
> We extend our sincere gratitude to the reviewer for their valuable time and insightful feedback. We value your constructive feedback and hope that our responses have appropriately addressed all the concerns.
>
> We really appreciate the valuable time to respond to our feedback based on the reviewer's comments. Further, we are happy to address any remaining concerns.

---

> > ### Comment · Reviewer_AFeH · 2024-12-03
> >
> > Thanks for the author's reply. I still have the following concerns:
> > 1. For t-sne visualization, I hope the author can add the latest method ArtPromptIR, because from Table 1, its method is the closest to the proposed method.
> > 2. There is a phenomenon that puzzles me. From Figure 2 and Figure 14, it can be seen that this work has shown a very outstanding decoupling ability for different degradation types. However, from the experiment in Table 1, it can be seen that the performance label of the proposed method is not very outstanding. This makes me question whether the good performance of this method is really due to the decoupling ability shown in t-sne? Because t-sne is just an analysis tool, it cannot directly explain that the effectiveness of the method is related to it.
> > 3. For some methods discussed by authors, InstructIR introduces a relatively new text prompt, and the work that is more similar should be PromptIR and TransWeather, both of which are relatively old. Therefore, I think the novelty of this work is still fair.

---

> > > ### Author Response · Authors · 2024-12-03
> > >
> > > We thank the reviewer for their time and valuable feedback and for the opportunity to address the remaining concerns.
> > >
> > > ``R1`` Unfortunately, the pre-trained weights of Art$_\text{PromptIR}$ are not publicly available, which prevents us from including it in the t-SNE plot.
> > >
> > > ``R2`` Correlation between t-SNE and quantitative performance:
> > > In all-in-one image restoration, a unified model is trained on a mix of degradation types. During inference, the model’s goal is to reason about the unspecified degradation type present in the input image and effectively remove it. AdaIR demonstrates strong reasoning and segregation capabilities, as evidenced by the t-SNE plots in Figures 2 and 14. While we agree that t-SNE is primarily a visualization tool and does not directly measure performance, it provides indirect validation of the model’s ability to capture degradation-specific features, which is in turn useful for effective restoration. The outstanding decoupling observed in the t-SNE visualizations is further supported by the quantitative results in Tables 1 and 2, where AdaIR achieves consistent and state-of-the-art performance across diverse tasks. These results collectively highlight AdaIR’s strength in addressing and adapting to varied degradation scenarios.
> > >
> > > ``R3`` We agree that TransWeather (CVPR 2022) and PromptIR (NeurIPS 2023) are from previous years; however, our method achieves better image quality scores even when compared to more recent works from 2024, such as Art$_\text{PromptIR}$ (ACM MM 2024), U-WADN, InstructIR (ECCV 2024), and GridFormer (IJCV 2024), as demonstrated in Tables 1 and 2.

---

### Official Review · Reviewer_FFqs · 2024-10-31

**Soundness:** 2
**Presentation:** 3
**Contribution:** 2
**Rating:** 5
**Confidence:** 4

**Summary:**

This paper proposes an adaptive all-in-one image restoration model called AdaIR using frequency information. A key motivation is that different degradation types impact the image’s different frequency sub-bands. AdaIR mines low- and high-frequency information from input features (FMiM) and modulates the extracted features by facilitating the cross-interaction between low-frequency mined features and high-frequency mined features (FMoM).

**Strengths:**

- The motivation for incorporating frequency information in an all-in-one image restoration model is well-founded. The authors substantiate their claim that various degradation types influence image content across frequency subbands. For instance, the t-SNE visualizations in Figures 2 and 11 illustrate how degradations can be clustered based on their types.
- The authors successfully demonstrate the function of each module within the model through extensive ablation studies. For example, in Figure 9, the visualizations of intermediate features and spectra effectively highlight the roles of the AFLB module, which is composed of the FiMM and FMoM components.

**Weaknesses:**

- The authors claim that all previous approaches operate solely in the spatial domain and do not consider frequency information (L086-L087). However, several studies [1-2] have already explored the frequency domain. The authors should clarify how their approach differs from or advances these existing frequency-based methods. Given the existence of prior frequency domain exploration, the primary motivation for this work may need to be revised.
- The proposed AdaIR appears to be a straightforward combination of existing techniques, such as Restormer [3] and CBAM [4]. Specifically, all encoders and decoders utilize multiple Transformer blocks from Restormer, and the cross attention (CA) mechanisms in both FMiM and FMoM leverage the MDTA approach from Restormer. Additionally, the H-L unit in FMoM seems to replicate the channel attention mechanism found in CBAM.
- The authors should appropriately cite the sources in both the figures and the main text. For example, in Figure 3, Restormer should be referenced for the TB in (a) and the CA in (d). Furthermore, CBAM should be cited since its channel attention mechanism is employed in the H-L unit of FMoM, both in Figure 3 and the main text. The authors should add citations to Restormer and CBAM in the caption of Figure 3, as well as in the relevant sections of the main text where these components are described.
- The roles of the H-L and L-H units appear to be similar in facilitating cross-interaction between low and high-frequency mined features. However, the authors have implemented different designs for these two units. It would be helpful for the authors to clarify the rationale behind using distinct methods for each.
- The metrics in the tables require clarification. For instance, Table 1 should include column headings that explicitly indicate the inclusion of PSNR and SSIM. The authors should add column headers in Table 1 (and any other relevant tables) to clearly indicate which metrics are being reported (e.g. "PSNR / SSIM").
- The benchmark models referenced are somewhat outdated. The authors should consider updating the comparison models to more recent ones [5-7]. Additionally, they should include comparisons with all-in-one restoration models that account for frequency information [1-2].


[1] Wen Y, Zhang K, Zhang J, Chen T, Liu L, Luo W. Frequency-oriented efficient transformer for all-in-one weather-degraded image restoration. IEEE Transactions on Circuits and Systems for Video Technology. 2023 Jul 27.

[2] Shi Z, Su T, Liu P, Wu Y, Zhang L, Wang M. Learning Frequency-Aware Dynamic Transformers for All-In-One Image Restoration. arXiv preprint arXiv:2407.01636. 2024 Jun 30.

[3] Zamir SW, Arora A, Khan S, Hayat M, Khan FS, Yang MH. Restormer: Efficient transformer for high-resolution image restoration. InProceedings of the IEEE/CVF conference on computer vision and pattern recognition 2022 (pp. 5728-5739).

[4] Woo S, Park J, Lee JY, Kweon IS. Cbam: Convolutional block attention module. InProceedings of the European conference on computer vision (ECCV) 2018 (pp. 3-19).

[5] Ai Y, Huang H, Zhou X, Wang J, He R. Multimodal Prompt Perceiver: Empower Adaptiveness Generalizability and Fidelity for All-in-One Image Restoration. InProceedings of the IEEE/CVF Conference on Computer Vision and Pattern Recognition 2024 (pp. 25432-25444).

[6] Park D, Lee BH, Chun SY. All-in-one image restoration for unknown degradations using adaptive discriminative filters for specific degradations. In2023 IEEE/CVF Conference on Computer Vision and Pattern Recognition (CVPR) 2023 Jun 17 (pp. 5815-5824). IEEE.

[7] Yang H, Pan L, Yang Y, Liang W. Language-driven All-in-one Adverse Weather Removal. InProceedings of the IEEE/CVF Conference on Computer Vision and Pattern Recognition 2024 (pp. 24902-24912).

**Questions:**

- Could the authors provide both the original and enlarged images in the figures? If this is not possible, assessing the performance of the proposed methods becomes challenging. For instance, while AdaIR appears to excel in rain removal, other models might perform better in different regions.
- The total number of parameters appears large (28.77M). Can authors add a column to their comparison tables (e.g. Tables 1 and 2) showing the number of parameters for each model? This would allow for a more comprehensive comparison of model efficiency alongside performance metrics.
- Can the authors clarify the term "Fixed" in Table 4? It is unclear whether the mask value is also fixed or not. The current caption in Table 4 only mentions that the mask shape is fixed at $10 \times 10$.
- Could the authors demonstrate that AdaIR, as an all-in-one model, performs well on unseen and mixed degradation using the UDC-SIT dataset [8], which features complex degradations?
- It would be helpful if the authors could explain more clearly the significance of the channel-wise features and the corresponding attention scores depicted in Figure 7.
- Can the authors elaborate on why AdaIR outperforms the Encoder + Decoder + AFLB configuration presented in Table 12? The explanation should include the advantages of AdaIR's design compared to the alternative.
- The additional ablation study regarding the design choices of FMoM in Appendix B may not be entirely fair. For example, the proposed design in Figure 8(b) does not initially concatenate $X_h$​ and $X_l$. Instead, each feature passes through the H-L or L-H unit before being cross-multiplied. The reviewer believes that the spatial attention comparison in Figure 8(a) should also avoid concatenation initially; instead, $X_h$​ and $X_l$​ should go through channel attention separately before their respective spatial attention maps are cross-multiplied. Could the authors provide a comparison based on this suggestion?

[8] Ahn, K., Ko, B., Lee, H., Park, C., & Lee, J. (2024). UDC-SIT: a real-world dataset for under-display cameras. Advances in Neural Information Processing Systems, 36.

**Details Of Ethics Concerns:**

- The H-L unit (L257-L268) appears to use the module from CBAM [4] without providing any citation.

[4] Woo S, Park J, Lee JY, Kweon IS. Cbam: Convolutional block attention module. InProceedings of the European conference on computer vision (ECCV) 2018 (pp. 3-19).

---

> ### Author Response · Authors · 2024-11-28
> **Response to Reviewer FFqs**
>
> ``R3.1``:*Frequency domain methods*:
>
> We appreciate the reviewer pointing out this oversight. We have now included frequency-based methods [1,2] in the paper and have reworded the relevant text accordingly. While methods [1,2] utilize manual or non-adaptive approaches for feature separation and perform frequency interactions without accounting for the distinct characteristics of different frequency components, our model introduces adaptive frequency separation. This enables the integration of frequency components based on the inherent properties of individual frequency subbands. The adaptability of frequency learning remains a core innovation of our work. To highlight this distinction, we have revised and refined the statements in both the abstract and introduction sections to enhance clarity and emphasize the unique contributions of our approach.
>
> ``R3.2``:*Reusage of existing techniques*:
>
> The primary goal of AdaIR is to tackle the challenge of adaptive all-in-one image restoration by introducing a frequency-guided learning framework. While AdaIR builds on foundational components from prior works (e.g., Transformers and attention mechanisms), its novel contributions lie in the macro-level integration of adaptive frequency learning through the proposed Adaptive Frequency Learning Block (AFLB). This block is specifically designed to decouple and modulate low- and high-frequency components in a way that is adaptive to the specific degradation present in the input image. Unlike prior frequency-based methods that employ manual or fixed approaches for feature separation, FMiM adaptively mines low- and high-frequency components through a mask generation block that dynamically adjusts to the spectral properties of the degradation. This adaptability ensures that frequency components are extracted in a degradation-specific manner. FMoM facilitates cross-band interaction by leveraging tailored attention mechanisms (H-L and L-H units) to exchange complementary information between frequency bands. This ensures that both global and local degradations are effectively addressed—a feature not seen in prior frequency-based models.
>
> Our objective is to develop a plug-in module tailored for all-in-one image restoration algorithms. To this end, we select the widely adopted Restormer as the baseline model. Additionally, to simplify the implementation of our mechanism and avoid extensive architecture tuning, we incorporate a CBAM-inspired attention mechanism. Despite the simplicity of design choices, our method achieves a remarkable performance improvement of 1.94 dB in PSNR compared to the baseline model.
>
> Our goal in this work was not to focus on introducing a multitude of fine-grained or incremental changes but rather to identify and implement simple yet key modifications. These targeted contributions allowed us to address the global objective of frequency-guided image restoration while ensuring the model remains computationally efficient and adaptable across multiple tasks.
>
> ``R3.3``:*Cite Restormer and CBAM works*:
>
>  In our initial submission, we credited Restormer while describing the overall pipeline in Section 3.1 and CBAM when comparing this mechanism with alternatives in Figure 8 and Appendix B (Design Choices of FMoM). Following the reviewer’s suggestion, we have added additional references for these two works in Figure 3 and the relevant sections of the main text where these components are described, ensuring proper acknowledgment and improved clarity.
>
> [1] Gao et al., Frequency-oriented efficient transformer for all-in-one weather-degraded image restoration, TCSVT 2023.
>
> [2] Shi et al., Learning Frequency-Aware Dynamic Transformers for All-In-One Image Restoration, arXiv 2024.

---

> ### Author Response · Authors · 2024-11-28
>
> ``R3.4``:*Rational behind different designs for H-L and L-H units*:
>
> High-frequency features are rich in spatial details, such as edges and textures. These features are transmitted to the low-frequency branch through a spatial attention mechanism, enabling it to focus on complex, localized regions. Conversely, low-frequency features provide global context, helping the high-frequency branch avoid overemphasizing challenging regions. To achieve this, we employ a channel attention mechanism that effectively integrates global information while assigning smaller weights to channels that overemphasize such regions, as demonstrated in Figure 7. The distinct designs are tailored to the characteristics of each frequency component. Spatial attention is well-suited for processing high-frequency features due to their localized nature, whereas channel attention is more effective for leveraging the global, structural information of low-frequency features. To demonstrate further the effectiveness of this design, we conducted additional experiments using alternative attention mechanisms for both units. The results, summarized in the table, show that our proposed design achieves the best performance, underscoring its effectiveness in facilitating cross-interaction between frequency components.
>
> **Table R1**: Comparisons between different attention methods
> |**Unit**|**Attention Type**|**PSNR**|
> |----|----|----|
> |(a) H-L/L-H |Channel/Channel| 30.10|
> |(b) H-L/L-H |Spatial/Spatial |30.36|
> |(c) H-L/L-H (Ours) |Spatial/Channel |30.52|
>
>
> ``R3.5``:*Explicit columns for PSNR/SSIM score*:
>
> Thank you for your suggestion. In response, we have explicitly added column headers to Tables 1, 2, and 12 to clearly indicate the metrics being reported, such as PSNR and SSIM.
>
> ``R3.6``:*Comparisons with recent methods*:
>
> Thank you for your valuable suggestion. As recommended, we have incorporated several recent methods, including U-WADN [3], Art$_{\text{PromptIR}}$ [4], Gridformer [5], and InstructIR [6], into the three-task and five-task all-in-one experiments. The updated results, provided in Table R2 and Table R3, demonstrate that our method performs competitively and achieves state-of-the-art results against these approaches.
>
> **Table R2** Results for the three-task setting (Table 1 in the paper)
>
> |**Method**|**Dehazing on SOTS**||**Deraining on Rain100L**||||**Denoising on BSD68**||||**Average**||Params|
> |---|---|---|---|---|---|---|---|---|---|---|---|---|---|
> ||||||$\sigma$=15||$\sigma$=25||$\sigma$=50|||||
> ||PSNR|SSIM|PSNR|SSIM|PSNR|SSIM|PSNR|SSIM|PSNR|SSIM|PSNR|SSIM|
> |Restormer| 7.78 |0.958 |33.78 |0.958 |33.72| 0.930| 30.67 |0.865 |27.63 |0.792| 30.75 |0.901 |26.13M|
> |PromptIR |30.58 |0.974 |36.37 |0.972| 33.98| 0.933 |31.31 |0.888| 28.06 |0.799 |32.06| 0.913 |32.96M|
> |U-WADN [3]|29.21 |0.971| 35.36 |0.968 |33.73 |0.931 |31.14| 0.886| 27.92| 0.793 |31.47 |0.910| 6M|
> |Art$_{\text{PromptIR}}$ [4] |30.83| 0.979 |37.94 |0.982| 34.06 |0.934 |31.42 |0.891 |28.14| 0.801 |32.49 |0.917| 33M|
> |Ours |31.06| 0.980| 38.64| 0.983| 34.12| 0.935 |31.45| 0.892| 28.19| 0.802| 32.69| 0.918 |28.77M|
>
> **Table R3** Results for the five-task setting (Table 2 in the paper)
>
> |**Method**|**Dehazing on SOTS**||**Deraining on Rain100L**||||**Denoising on BSD68**||||**Average**||Params|
> |---|---|---|---|---|---|---|---|---|---|---|---|---|---|
> ||||||$\sigma$=15||$\sigma$=25||$\sigma$=50|||||
> ||PSNR|SSIM|PSNR|SSIM|PSNR|SSIM|PSNR|SSIM|PSNR|SSIM|PSNR|SSIM|
> |PromptIR |26.54 |0.949 |36.37 |0.970 |31.47| 0.886 |28.71| 0.881| 22.68 |0.832| 29.15| 0.904 |32.96M|
> |Gridformer[5] |26.79| 0.951 |36.61 |0.971 |31.45 |0.885 |29.22| 0.884 |22.59 |0.831| 29.33 |0.904| 34.07M|
> |InstructIR[6] |27.10| 0.956| 36.84| 0.973| 31.40 |0.887 |29.40 |0.886 |23.00 |0.836| 29.55| 0.907 |15.80M|
> |Ours |30.53 |0.978| 38.02 |0.981 |31.35 |0.889 |28.12 |0.858 |23.00| 0.845 |30.20| 0.910 |28.77M|
>
> [3] Xu et al., Unified-width adaptive dynamic network for all-in-one image restoration, arXiv 2024.
>
> [4] Wu et al., Harmony in diversity: Improving all-in-one image restoration via multi-task collaboration, ACMMM 2024.
>
> [5] Wang et al.,  Gridformer: Residual dense transformer with grid structure for image restoration in adverse weather conditions, IJCV 2024.
>
> [6] Conde et al., High-quality image restoration following human instructions, ECCV 2024.
>
> ``R3.7``:*Visualizations*:
>
> We have included both the original and enlarged versions of the images in Figures 4–6 for better visualization. Our method is more effective in removing different degradations.
>
> ``R3.8``:*Parameter comparisons*:
>
> Parameter comparisons have been incorporated into Tables 1 and 2. Our method performs favorably against state-of-the-art algorithms while maintaining a comparable parameter count.
>
> ``R3.9``:*Clarification for fixed masks*:
>
> The term *fixed* indicates that a $10\times 10$ mask is utilized in our experiment, and this value remains constant throughout the entirety of the experiment.

---

> ### Author Response · Authors · 2024-11-28
>
> ``R3.10``:*Generalization on unseen datasets*:
>
> Please note, as with previous works like AirNet and PromptIR, all-in-one models are designed to address unspecified degradation types rather than unseen corruptions. In this context, “multiple degradations” refers to datasets that contain a variety of degradation types, where each image exhibits only one specific type of degradation. Consequently, a model trained on a limited set of degradation types cannot be expected to perform optimally on all unseen complex degradations.
>
> Achieving universal applicability to all possible degradation types, including unseen and mixed degradations, remains a highly challenging research goal. While it would be ideal to address such universal scenarios, it is beyond the current state-of-the-art capabilities of all-in-one restoration models, including AdaIR.
>
> ``R3.11``:*Channel attention scores*:
>
> Given that low-frequency features offer a global perspective, we employ a channel attention mechanism in the L-H unit to transfer low-frequency features to the high-frequency branch. This approach helps high-frequency features avoid overemphasizing challenging regions. As illustrated in Figure 7, the channel attention unit assigns smaller weights to channels that overly focus on challenging regions, while allocating larger weights to other channels. This effectively reinforces the already recovered sharper regions, contributing to improved restoration performance.
>
> ``R3.12``:*Clarification for Encoder+Decoder+AFLB*:
>
> The encoder is responsible for extracting deep features from the input, while the decoder focuses on reconstructing sharp, high-resolution features. During the encoding stage, the model may struggle to provide meaningful high- and low-frequency information for effective exchange, which can adversely affect overall performance. A similar observation has also been reported in PromptIR.
>
> ``R3.13``:*Additional ablation studies of using separate attention*:
>
> Following the suggestion, we separately process low- and high-frequency features using channel attention mechanisms. The results presented in Table R1a and Table R1c demonstrate that our model achieves superior performance under this configuration. Additionally, we evaluated the use of separate spatial attention mechanisms in the two branches (Table R1b), which resulted in lower performance compared to our proposed approach.
>
> ``R3.14``:*Citation for CBAM*:
>
> In the original paper, we cited this method in Appendix B (Design Choices of FMoM) and Figure 8. Following the suggestion, we have now ensured that this citation is included comprehensively throughout the manuscript wherever relevant.

---

> ### Author Response · Authors · 2024-12-02
> **We thank the reviewer again for the valuable feedback and happy to address any remaining concerns.**
>
> We extend our sincere gratitude to the reviewer for their valuable time and insightful feedback. We value your constructive feedback and hope that our responses have appropriately addressed all the concerns.
>
> We really appreciate the valuable time to respond to our feedback based on the reviewer's comments. Further, we are happy to address any remaining concerns.

---

> ### Comment · Reviewer_FFqs · 2024-12-03
>
> I appreciate the authors' efforts in addressing the concerns raised. After reviewing the authors' responses and discussions with other reviewers, many issues have been resolved, and I have raised my score to 5. However, this does not indicate support for acceptance of this paper but reflects that the paper does not warrant rejection at a score of 3.
>
> **1. Lack of comparisons with frequency-based models**
>
> The authors did not provide the performance comparisons with frequency-based models [1-2] requested in the initial review. Although new experiments cannot be requested at this stage, I reiterate this point as the benchmark comparisons are crucial to validating the authors' stated motivation for this work.
>
> The absence of experiments comparing the proposed method with frequency-based all-in-one restoration models would make it difficult to evaluate whether the proposed method is better than the previous frequency-based models, as the authors reply in R3.1.
>
> I surveyed the performance of previous work [2] and compared it with AdaIR. AdaIR shows slightly lower performance (PSNR: -0.37, SSIM: -0.009) on BSD68 compared to previous work, suggesting the need for direct comparisons with similar approaches to validate the novelty.
>
> **2. Novelty and insufficient analysis of MGB**
>
> The MGB is said to adaptively mine frequency components, but no analysis shows how and why it is superior to existing methods like FFTformer [3], which uses a Quantization Matrix for frequency selection. Extracting important frequency components has been addressed in prior works, and aside from this, this is the only module that offers substantial novelty.
>
> **3. Citation concerns with the H-L unit**
>
> The authors renamed CBAM to the H-L unit without citation in the main text, which could mislead readers into thinking the H-L unit is newly proposed. While the authors added citations in the appendix, the main text must clarify this, e.g., "We use CBAM's spatial attention as the H-L unit." Proper attribution in the main text is essential to avoid misunderstanding.
>
> [1] Wen Y, Zhang K, Zhang J, Chen T, Liu L, Luo W. Frequency-oriented efficient transformer for all-in-one weather-degraded image restoration. IEEE Transactions on Circuits and Systems for Video Technology. 2023 Jul 27.
>
> [2] Shi, Zenglin, et al. "Learning Frequency-Aware Dynamic Transformers for All-In-One Image Restoration." arXiv preprint arXiv:2407.01636 (2024).
>
> [3] Kong, Lingshun, et al. "Efficient frequency domain-based transformers for high-quality image deblurring." CVPR 2023.

---

> > ### Author Response · Authors · 2024-12-04
> >
> > ``R1`` Our method consistently achieves better image fidelity scores compared to several recent state-of-the-art models introduced in 2024, including Art$_\text{PromptIR}$ (ACM MM 2024), U-WADN, InstructIR (ECCV 2024), and GridFormer (IJCV 2024), as demonstrated in Tables 1 and 2.
> >
> > For frequency-based methods, our model employs adaptive frequency decoupling, whereas prior approaches rely on manual or non-adaptive techniques. Moreover, our method explicitly models interactions between different frequency bands by accounting for their distinct characteristics, in contrast to earlier works that use frequency features as direct inputs to Transformer blocks without such considerations.
> >
> > In terms of quantitative performance, the proposed AdaIR achieves an average PSNR of 32.69 dB across all datasets, surpassing the previous work [2] by 0.37 dB. Notably, it delivers significant improvements in specific tasks, outperforming the algorithm of [2] by 1.86 dB in dehazing and 1.14 dB in deraining. (Please note that the work [2] is an Arxiv preprint and not a peer-reviewed published work, yet.)
> >
> > ``R2`` FFTformer employs a quantization matrix, which is a learnable but fixed matrix after training, for frequency selection in image deblurring. However, this design lacks the flexibility to adapt to varying degradations and does not support interactions between different frequency subbands.
> >
> > In contrast, our method dynamically adapts to diverse inputs across multiple tasks and facilitates effective interactions between frequency subbands, providing a more robust and versatile solution. To further demonstrate the advantages of our approach, we conducted experiments by incorporating a quantization matrix within our framework to modulate features at the same locations. This alternative design resulted in a PSNR that was 0.86 dB lower than our proposed method, highlighting the benefits of our adaptive frequency design. (Please note that this ablation experiment was conducted for 90K iterations due to time constraints.)
> >
> > ``R3`` Thank you. We utilize two distinct units for cross-communication: specifically, H-L to deliver the decoupled high-frequency features to the low-frequency branch, and L-H for the opposite direction. This is the rationale behind the naming of H-L and L-H. We will incorporate the phrase suggested by the reviewer in the camera-ready version.

---

### Official Review · Reviewer_K6x2 · 2024-11-02

**Soundness:** 4
**Presentation:** 3
**Contribution:** 3
**Rating:** 8
**Confidence:** 5

**Summary:**

This paper presents an all-in-one image restoration framework that can generate clean images from multiple degradation patterns, such as noise, blur, haze, and rain. Different from existing all-in-one algorithms that perform only in the spatial domain, the proposed network deals with all-in-one tasks from the perspectives of frequency mining and modulation. The network extracts different frequencies from the degradation images and modulates features for high-quality reconstruction. The adaptive strategy enhances the learning of informative frequency signals based on input degradation. Comprehensive experiments on two kinds of all-in-one settings demonstrate the model's effectiveness.

**Strengths:**

1) The paper presents a novel adaptive all-in-one image restoration architecture that can effectively decouple the input into different frequencies and modulate them for high-quality reconstruction. The paper is well-motivated, and Figure 1 makes sense.
2) The proposed blocks are reasonable and can be easily injected into other algorithms.
3) The visualizations well demonstrate the efficacy of the proposed model and motivation.
4) The model is computationally efficient. It achieves sota, and the experiments on two kinds of all-in-one settings are thorough, while other algorithms only experiment on a single setting.

**Weaknesses:**

1) The comparisons using only PSNR/SSIM seem insufficient to measure the quality of the resulting images. LPIPs can be used to further verify the proposed model's superiority.
2) The architecture is mainly based on Restormer. However, Tab. 1 does not include Restormer's performance.
3) The paper lacks comparisons with recent methods, such as Harmony in Diversity: Improving All-in-One Image Restoration via Multi-Task Collaboration.

**Questions:**

Please refer to the weaknesses.

---

> ### Author Response · Authors · 2024-11-28
> **Response to Reviewer K6x2**
>
> ``R2.1``:*LPIPS evaluation*:
>
> As recommended, we assess the image reconstruction quality of our method against state-of-the-art approaches using the LPIPS metric. The results presented in the table indicate that our method surpasses the performance of existing approaches.
>
> |**Method**|**Dehazing on SOTS**|**Deraining on Rain100L**||**Denoising on BSD68**||**Average**|
> |---|---|---|---|---|---|---|
> ||||$\sigma$=15|$\sigma$=25|$\sigma$=50||
> |AirNet|0.0264 |0.0277 |0.0642| 0.1142| 0.2119 |0.0889|
> |PromptIR|0.0132| 0.0162| 0.0639 |0.1132| 0.2144| 0.0842|
> |Ours|0.0116 |0.0118 |0.0611| 0.1079| 0.2112 |0.0807|
>
> ``R2.2``:*Baseline performance*:
>
> As recommended, we have incorporated Restormer into our comparative analysis. Notably, our proposed method demonstrates a substantial performance improvement, achieving a PSNR gain of 1.94 dB over the baseline model.
>
> |**Method**|**Dehazing on SOTS**||**Deraining on Rain100L**||||**Denoising on BSD68**||||**Average**||
> |---|---|---|---|---|---|---|---|---|---|---|---|---|
> ||||||$\sigma$=15||$\sigma$=25||$\sigma$=50||||
> ||PSNR|SSIM|PSNR|SSIM|PSNR|SSIM|PSNR|SSIM|PSNR|SSIM|PSNR|SSIM|
> |Restormer |27.78 |0.958| 33.78 |0.958 |33.72| 0.930| 30.67 |0.865 |27.63| 0.792 |30.75 |0.901|
> |Ours |31.06| 0.980 |38.64 |0.983 |34.12 |0.935 |31.45 |0.892 |28.19| 0.802 |32.69|0.918|
>
> ``R2.3``:*Comparisons with more recent methods*:
>
> Thank you for pointing out this paper. Compared with Art$_{\text{PromptIR}}$ [1], our method achieves an average performance improvement of 0.2 dB. We have included this work, along with several other recent methods, in our comparisons. Please refer to Tables 1 and 2 in the main paper for the updated results.
>
> |**Method**|**Dehazing on SOTS**||**Deraining on Rain100L**||||**Denoising on BSD68**||||**Average**||
> |---|---|---|---|---|---|---|---|---|---|---|---|---|
> ||||||$\sigma$=15||$\sigma$=25||$\sigma$=50||||
> ||PSNR|SSIM|PSNR|SSIM|PSNR|SSIM|PSNR|SSIM|PSNR|SSIM|PSNR|SSIM|
> |Art$_{\text{PromptIR}}$| 30.83| 0.979| 37.94 |0.982 |34.06| 0.934 |31.42| 0.891 |28.14 |0.801 |32.49| 0.917|
> |Ours |31.06 |0.980 |38.64| 0.983 |34.12| 0.935 |31.45| 0.892| 28.19 |0.802 |32.69| 0.918|
>
> [1] Wu et al., Harmony in diversity: Improving all-in-one image restoration via multi-task collaboration, ACMMM 2024.

---

> ### Author Response · Authors · 2024-12-02
> **We thank the reviewer again for the valuable feedback and happy to address any remaining concerns.**
>
> We extend our sincere gratitude to the reviewer for their valuable time and insightful feedback. We value your constructive feedback and hope that our responses have appropriately addressed all the concerns.
>
> We really appreciate the valuable time to respond to our feedback based on the reviewer's comments. Further, we are happy to address any remaining concerns.

---

> > ### Comment · Reviewer_K6x2 · 2024-12-02
> > **Thank you for your reply.**
> >
> > Thank you for your reply. After reading your response and the other reviewers' comments, I decided to maintain my rating to support the acceptance of this paper.

---

### Official Review · Reviewer_Jdae · 2024-11-09

**Soundness:** 3
**Presentation:** 3
**Contribution:** 3
**Rating:** 6
**Confidence:** 4

**Summary:**

The paper presents AdaIR, which aims to address multiple forms of degradation, such as noise, blur, haze, and rain, within a single model. AdaIR is motivated by the observation that different types of degradation impact image content on different frequency subbands.  AdaIR leverages both spatial and frequency domain information to effectively separate and remove these degradations using customised modules.

**Strengths:**

The paper highlights a key insight for any image restoration: Noisy and rainy images are contaminated with high-frequency content while the low-frequency content degrades more in Low-light and hazy images.

Exploiting frequency domain is a key differentiation, as previous methods like AirNet, IDR, and PromptIR operate solely on spatial domain information

Frequency Mining Module (FMiM) extracts relevant frequency signals from degraded images through adaptive spectral decomposition, targeting degradation-specific frequency content. The use of a dynamic, learnable Mask Generation Block (MGB) to separate low- and high-frequency representations based on the input image's degradation type allows AdaIR to adapt to different degradations effectively

Frequency Modulation Module (FMoM) refines these frequency features by enabling interaction between different frequency components, enhancing the model’s effectiveness on various degradation types.

Experiments on a wide set of benchmarks demonstrates consistent performance gains over competing approaches in both three-degradation and five-degradation settings

The improved quality of discriminative feature learning (shown in Fig. 2) is impressive.

**Weaknesses:**

While the paper demonstrates the generalisation ability of AdaIR by testing it on an unseen desnowing task and synthetic images with mixed degradations (rain and noise) in Table 7 and 8, there are no experiments on real out-of-distribution images. Extensive qualitative comparisons on real images should be included. Evaluation on a wider range of out-of-distribution degradations would strengthen the claims of the model's generalisation capabilities.

The quantitative tables primarily compare AdaIR with other all-in-one methods but not the scores of single-task trained state-of-the-art models. Including the highest scores achieved by SotA single task models on each individual task would provide a more comprehensive assessment of AdaIR's performance relative to specialised approaches.

While Table 4 shows the overhead of AFLB in terms of parameters and FLOPs, a more in-depth analysis of its computational cost and potential impact on real-time applications would be beneficial.
FLOPs are only reported for ablation studies, but there are no FLOPs comparisons with competing methods.
In addition to FLOPs, there should also be comparison between inference time of different models.
using frequency domain processing inherently introduces additional processing for transformations eg IFFT.
Although they are beneficial for accuracy, a potential drawbacks of such approaches is the computation cost and incompatibility with edge platforms.

The paper doesn't cite or compare with any existing models that use frequency domain processing. There have been many such studies [refs 1-7 below]. Some of these papers are quite related as they propose transformer modules that focus on high/low frequencies.

Furthermore, comparing with alternative transform-domain methods, such as wavelet transforms or other adaptive filtering techniques, commonly used in related papers, could enhance the performance and adaptability of AdaIR.

The MGB dynamically generates a mask to separate low and high frequencies based on the input image. It would be interesting to see the difference in generated masks for different tasks. This vanilla approach could be prone to overfitting, especially since the training dataset does not adequately cover the diversity of real-world degradations. A more robust approach might involve specifically incorporating prior knowledge about frequency characteristics of different degradations or exploring regularization techniques to prevent overfitting

Paper lacks deeper analysis on how the interaction between different frequency bands contributes to the restoration process. Such analysis could lead to more informed design choices and potentially improve the interpretability of the model.

The graph in Fig. 1 right doesn’t clearly demonstrate the stark frequency-domain differences between different tasks.


[1] Seeing the Unseen: A Frequency Prompt Guided Transformer for Image Restoration (ECCV 2024)

[2] Selective Frequency Network for Image Restoration (ICLR 2023)

[3] Image Restoration via Frequency Selection, TPAMI 2024

[4] Intriguing Findings of Frequency Selection for Image Deblurring, AAAI 2023

[5] Efficient Frequency Domain-based Transformer for High-Quality Image Deblurring, CVPR 2023

[6] LoFormer: Local Frequency Transformer for Image Deblurring [ACM MM 2024]

[7] Learning Frequency Domain Priors for Image Demoireing, TPAMI 2022

**Questions:**

Address weaknesses above.

---

> ### Author Response · Authors · 2024-11-28
> **Response to Reviewer Jdae**
>
> ``R1.1a``:*Out-of-distribution generalization experiments*:
>
> As suggested, we assess the generalization capability of our model on additional out-of-distribution degradation types: out-of-focus blurring and raindrops. Results show that our method demonstrates superior performance on these unseen degradation types compared to other competing approaches. (These results are also provided in Table 9 of the paper.)
> ||**DPDD** [1]||**AGAN** [2]||
> |----|----|----|----|----|
> |**Method**|PSNR|SSIM|PSNR|SSIM|
> |AirNet|20.17 |0.662 |22.09| 0.822|
> |PromptIR |21.76 |0.661| 22.98| 0.827|
> |Ours |22.93| 0.711 |23.14 |0.826|
>
>
> ``R1.1b``:*Real-image visual example*:
>
> To provide qualitative results on real-world images, we use the UAVDT [3] dataset, which consists of images captured by UAVs at varying altitudes and exhibiting diverse levels of haze. Visual comparisons are provided in Figure 8 of the paper.
>
> ``R1.2``:*Compare AdaIR with Specialized single-task approaches*:
>
> We appreciate your suggestion to compare AdaIR’s performance with specialized single-task models for a more comprehensive evaluation. However, we would like to clarify the following points. (1) SOTA single-task models are typically designed and optimized for handling a specific type of degradation. As a result, they inherently benefit from task-specific architectures, making them more capable of achieving superior performance compared to all-in-one architectures. (2) To ensure a fair and consistent evaluation, we followed the experimental protocols established by prior works such as PromptIR and AirNet. This allows us to avoid retraining all the competing methods and simply extend existing quantitative tables. (3) All-in-one methods (PromptIR, AirNet, AdaIR, etc.) use much smaller training datasets compared to single-task models. Comparing AdaIR directly with single-task SotA models would require retraining these specialized models on the same dataset used for AdaIR. Unfortunately, this is not feasible within the limited time and computing resources we have.
>
> ``R1.3``:*Computational comparisons*:
>
> We have incorporated FLOPs comparisons in Table 17. Notably, despite the additional processing introduced by Fourier transformations, our model demonstrates competitive computational efficiency. Specifically, AdaIR operates 1.28x faster than the PromptIR approach, while achieving a significant performance improvement of 0.63 dB in PSNR and utilizing 19% fewer parameters.
>
> We acknowledge the potential concerns regarding compatibility with edge platforms. However, this challenge is not unique to AdaIR and applies broadly to all the compared approaches. For deployment on edge devices, further optimization would be necessary to reduce AdaIR’s computational requirements, which could be achieved by reducing model capacity.
>
> ``R1.4``:*Additional references*:
>
> We thank the reviewer for providing valuable references to related works. These references have now been included in the Related Work section for a more comprehensive discussion.
>
> ``R1.5``:*Comparison with adaptive-filtering approach*:
>
> As suggested, we conduct experiments using an adaptive method for frequency decoupling [4]. This variant achieves a PSNR of 30.25 dB, which is 0.99 dB lower than the performance of our AdaIR algorithm.
>
> ``R1.6``:*Masks for different tasks*:
>
> We have included visualizations for additional degradation types in Figure 11 of the paper. As seen, our model can adaptively decouple the images into different frequency bands guided by the generated masks.
>
> ``R1.7``:*How the interaction contributes to restoration*:
>
> Figure 7 demonstrates the complementary relationship between different frequency bands. Specifically, high-frequency features aid the low-frequency branch by emphasizing spatial edges, while low-frequency features assist the high-frequency branch in avoiding excessive emphasis on challenging areas. Consequently, the combined features enhance the sharpness of the input features, facilitating high-quality reconstruction, as depicted in Figure 12.
>
> ``R1.8``:*Frequency differences*:
>
> Figure 1 illustrates that the amplitude variations across tasks highlight how different degradations affect images at distinct frequency ranges.
>
> [1] Abuolaim et al., Defocus deblurring using dual-pixel data, ECCV 2020.
>
> [2] Qian et al., Attentive generative adversarial network for raindrop removal from a single image, CVPR 2018.
>
> [3] Du et al., The unmanned aerial vehicle benchmark: Object detection and tracking, ECCV 2018.
>
> [4] Zhou et al., Seeing the unseen: A frequency prompt guided transformer for image restoration, ECCV 2024.

---

> ### Author Response · Authors · 2024-12-02
> **We thank the reviewer again for the valuable feedback and happy to address any remaining concerns.**
>
> We extend our sincere gratitude to the reviewer for their valuable time and insightful feedback. We value your constructive feedback and hope that our responses have appropriately addressed all the concerns.
>
> We really appreciate the valuable time to respond to our feedback based on the reviewer's comments. Further, we are happy to address any remaining concerns.

---

### Meta-Review · Area_Chair_U3b1 · 2024-12-20

**Metareview:**

All reviewers agree on the innovation and novelty of the proposed work. Reviewers had individual questions regarding the presentation of the results, which the authors responded rigorously. This led to reviewers increasing their initial score after the rebuttal to the average score of 6.
Therefore the paper is recommended for acceptance.

**Additional Comments On Reviewer Discussion:**

No discussion between the reviewers,

---

### Decision · Program_Chairs · 2025-01-22

Accept (Poster)